# HeurAgenix: A Multi-Agent LLM-Based Paradigm for Adaptive Heuristic Evolution and Selection in Combinatorial Optimization

## Abstract

Combinatorial Optimization (CO) is a class of problems where the goal is to identify an optimal solution from a finite set of feasible solutions under specific constraints. Despite its ubiquity across industries, existing heuristic algorithms struggle with limited adaptability, complex parameter tuning, and limited generalization to novel problems. Recent approaches leveraging machine learning have made incremental improvements but remain constrained by extensive data requirements and reliance on historical problem-specific adjustments. Large Language Models (LLMs) offer a new paradigm to overcome these limitations due to their ability to generalize across domains, autonomously generate novel insights, and adapt dynamically to different problem contexts. To harness these capabilities, we introduce **HeurAgenix**, a novel multi-agent hyper-heuristic framework that leverages LLMs to generate, evolve, evaluate, and select heuristics for solving CO problems. Our framework comprises four key agents: heuristic generation, heuristic evolution, benchmark evaluation, and heuristic selection. Each agent is designed to exploit specific strengths of LLMs, such as their capacity for synthesizing knowledge from diverse sources, autonomous decision-making, and adaptability to new problem instances. Experiments on both classic and novel CO tasks show that HeurAgenix significantly outperforms state-of-the-art approaches by enabling scalable, adaptable, and data-efficient solutions to complex optimization challenges.

## 1 Introduction

Combinatorial Optimization (CO) problems are fundamental to many disciplines, ranging from production scheduling and resource allocation to finance and energy management. These problems require finding optimal solutions from a discrete set of possibilities while adhering to predefined constraints. Traditional algorithms, particularly exact methods, are limited to small-scale problems due to their computational complexity. In contrast, heuristic methods, although more scalable, often face issues such as limited adaptability, difficult parameter tuning, and limited generalization across problem domains. The manual effort required to fine-tune heuristics for each new problem instance is a significant bottleneck (Peres & Castelli, 2021).

In recent years, hyper-heuristic approaches have attempted to bridge this gap by automating the selection or generation of heuristics based on problem characteristics. These methods include adaptive selection hyper-heuristics (Drake et al., 2020), genetic programming-based heuristic generation (Nguyen et al., 2011), and iterative local search techniques (Burke et al., 2010). While these approaches enhance generalization, they still struggle with domain-specific sensitivity, requiring extensive testing and adjustment. Karimi-Mamaghan et al. (2022) and Mahendran et al. (2020) have incrementally enhanced these methods with machine learning-based improvements, but challenges such as data dependency, overfitting, and scalability remain.

Large Language Models (LLMs) offer a transformative leap forward in solving these shortcomings. Unlike traditional approaches that rely on domain-specific heuristics or rigid algorithms, LLMs possess several unique capabilities that make them well-suited for CO problems:

- **Generalization across domains:** LLMs are pre-trained on diverse corpora, enabling them to understand and apply knowledge across various problem types without the need for extensive domain-specific fine-tuning.

- **Autonomous knowledge synthesis:** LLMs can generate novel heuristics by combining internal knowledge with external references, allowing them to propose creative, previously unexplored solutions.

- **Adaptability to dynamic environments:** LLMs can rapidly adapt to new problem instances by generating solutions informed by the specific context of the problem, making them highly versatile in handling evolving or unseen CO tasks.

- **Efficient decision-making through abstraction:** LLMs excel at abstract reasoning, allowing them to decompose complex optimization problems and propose solutions that balance immediate gains with future improvements.

These capabilities, when applied to CO, can significantly reduce the need for manual intervention, extensive data requirements, and problem-specific tuning, providing a more scalable and robust solution to complex optimization problems. Despite the potential of LLMs, existing applications of LLMs in CO have several limitations. Previous studies such as FunSearch(Romera-Paredes et al., 2024), EoH(Liu et al., 2024a), and ReEvo(Ye et al., 2024) have successfully leveraged LLMs for heuristic generation and evolutionary search. However, these approaches still rely heavily on existing approaches. Moreover, they often follow rigid, single-agent architectures where each heuristic operates in isolation, limiting the system's ability to adapt dynamically to new and complex problem instances.

**To address these limitations, we propose HeurAgenix, a multi-agent hyper-heuristic framework that fully integrates LLMs across all stages of CO problem-solving.** Unlike previous approaches, HeurAgenix deploys a multi-agent system that leverages the specific strengths of LLMs for different stages of heuristic management, as follows:

- **Heuristic Generation Agent:** This agent capitalizes on the LLMs' ability to generate heuristics from multiple sources, including internal knowledge, reference papers, and related problem heuristics. By synthesizing diverse knowledge, the agent generates novel and adaptive heuristics tailored to a wide variety of CO tasks.

- **Heuristic Evolution Agent:** Using LLMs' capabilities for autonomous decision-making and reflection, this agent evolves heuristics by comparing multiple solutions, identifying bottlenecks, and iteratively refining the heuristics based on performance data without relying on human domain knowledge.

- **Benchmark Evaluation Agent:** LLMs' abstract reasoning allows this agent to develop comprehensive feature extractors that characterize both the problem instance and the current solution. This enables deeper insights into the problem, allowing for more informed decision-making during the optimization process.

- **Heuristic Selection Agent:** LLMs' capacity for dynamic decision-making enables this agent to choose the most appropriate heuristic based on real-time evaluation of features. This ensures robust performance across different problem instances and states, dynamically adapting to changes as the problem evolves.

By leveraging the full suite of LLM capabilities, our multi-agent framework not only automates heuristic design but also provides a highly adaptable, scalable solution to a wide range of CO problems. Extensive experiments on classical problems such as the Traveling Salesman Problem (TSP) and novel challenges like the Dynamic Production Order Scheduling Problem (DPOSP) demonstrate that HeurAgenix significantly outperforms existing approaches in terms of adaptability, performance, and scalability. We will make all the codes publicly available upon the publication of our paper.

## 2 RELATED WORK

**Generative Hyper-Heuristics** Generative hyper-heuristics are techniques that automatically generate new heuristics by amalgamating elementary operations or decision-making rules, such as genetic programming, genetic algorithms, and particle swarm optimization (Hou et al., 2023; Singh & Pillay,

2022). However, generative hyper-heuristics face challenges such as high computational load, parameter tuning complexity, and limited adaptability. To address these issues, contemporary research has been concentrating on integrating of deep learning techniques, and the development of adaptive heuristic generation strategies. These advancements aim to significantly enhance the adaptability, efficiency, and overall performance of generative hyper-heuristics (Jia et al., 2019; Wu et al., 2021).

**Selection Hyper-Heuristics**    Selection hyper-heuristics optimize by selecting the most suitable heuristic from a predefined set to adapt to the current problem scenario. These algorithms typically employ rule-based selection, meta-heuristic selection, or learning-based selection methods, making them well-suited for dynamic optimization problems and complex combinatorial scenarios (de Carvalho et al., 2021; Drake et al., 2020). However, selection hyper-heuristics face challenges such as complex selection strategies, reliance on historical data, and limited generalization ability. Recent advancements aim to improve robustness and adaptability by incorporating reinforcement learning to enhance selection strategies, exploring online learning methods, and developing hybrid selection techniques that effectively combine multiple strategies (de Santiago Junior et al., 2020; Sopov, 2016).

**LLMs for Combinatorial Optimization**    LLMs have demonstrated significant potential in various domains, including CO. Zhang et al. (2024) evaluated the performance of current LLMs on various graph optimization problems. Iklassov et al. (2024) designed effective prompt strategies to address CO issues. Xiao et al. (2023) introduced the Chain-of-Experts approach, leveraging multi-agent cooperation to directly solve optimization problems.

More relevant to our work are studies leveraging LLMs to generate and evolve heuristic algorithms for solving CO problems. Romera-Paredes et al. (2024) introduced FunSearch, a novel approach that utilizes LLMs to evolve heuristics for CO problems. EoH (Liu et al., 2024a) advances FunSearch by introducing multi-directional evolution to increase the diversity of heuristic algorithms. ReEvo (Ye et al., 2024) further refines this process by integrating LLM-driven reflection, enhancing the efficiency of the evolution of heuristics. These works have significantly improved the effectiveness of heuristics by leveraging the strengths of LLMs. However, these approaches still rely on expert knowledge and manual design, and thus, they cannot directly yield end-to-end solutions, especially when addressing novel problems.

As illustrated in Table 1, our HeurAgenix approach introduces key innovations to tackle these issues. These include integrating multiple sources (LLMs' internal knowledge, reference papers, and related problems) for heuristic generation, employing a data-driven approach for heuristic evolution, and using LLM-generated features for evaluation and heuristic selection to ensure robust performance across diverse problems.

Table 1: Comparison of LLM-based CO paradigms on heuristic generation, evolution, evaluation and selection.

| Paradigm | Heuristic generation | Heuristic evolution | Benchmark evaluation | Heuristic selection |
|---|---|---|---|---|
| FunSearch | Generation from LLM | Single-direction evolution | Manual design metrics | Manual design strategies |
| EoH | Generation from LLM | Multiple-directions evolution | Manual design metrics | Manual design strategies |
| ReEvo | Generation from LLM | Feedback-guided evolution | Manual design metrics | Manual design strategies |
| HeurAgenix (Ours) | Generation from multiple sources | Data-driven evolution | LLM-generated feature | Feature-based LLM selection |

## 3    METHODOLOGY

As depicted in Figure 1, HeurAgenix operates through two main phases to solve CO problems. In the heuristic generation phase, the **heuristic generation agent** generates heuristics from LLM's internal knowledge, reference papers, or related problems' heuristics, while the **heuristic evolution agent** evolves these heuristics using training data. During the problem solving phase, the **benchmark evaluation agent** generates feature extractors for the problem instance and solution, and the **heuristic selection agent** dynamically selects the appropriate heuristic based on these features.

### 3.1    HEURISTIC GENERATION PHASE

In this paper, the heuristic is represented as the function $H : H(G, S, P) \to S'$, where $G$ is the instance data, $S$ is the current (partial) feasible solution, and $P$ consists of all heuristic parameters.

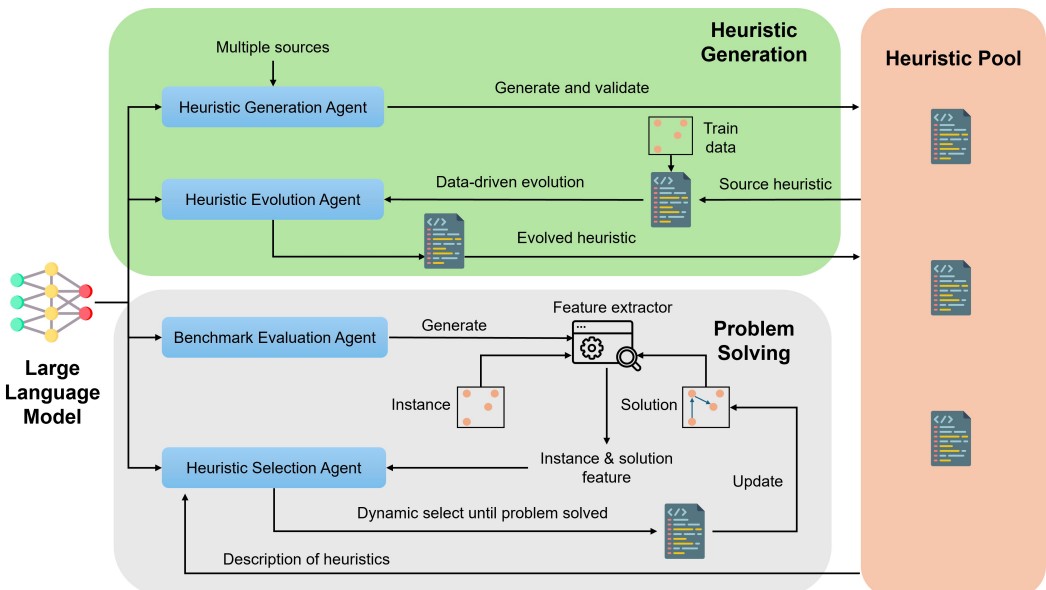

Figure 1: The framework and agents of HeurAgenix.

The function $H$ yields a new solution state $S'$ through a single-step operation such as addition, deletion, replacement, exchange, or perturbation, ensuring the search process is controlled (Hillier & Lieberman, 2015).

### 3.1.1 HEURISTIC GENERATION AGENT

Due to a phenomenon known as hallucinations, directly using LLMs to generate heuristics for new problems often leads to incorrect heuristics (Mündler et al., 2024). As illustrated in Figure 2, to reduce hallucinations, the heuristic generation agent learns from multiple sources and employs a smoke test to ensure the correctness of the generated heuristics.

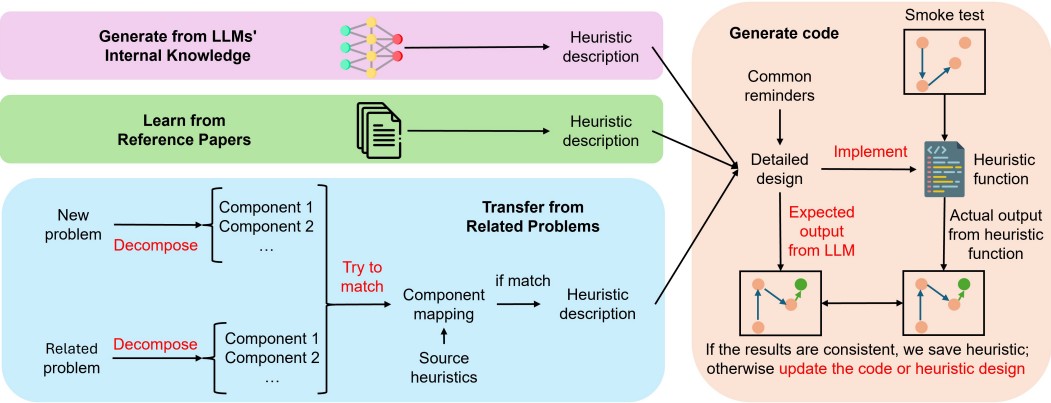

Figure 2: The heuristic generation process. The red text indicates interactions with the LLM.

Heuristics can be generated directly from **LLM's internal knowledge**. A similar approach has been adopted by Funsearch (Romera-Paredes et al., 2024), EoH (Liu et al., 2024a), and ReEvo (Ye et al., 2024) to obtain initial heuristics. Besides, we can also learn heuristics from **reference papers**. The LLM first reads the abstract to determine relevance, then selects interesting sections, and finally decides whether to generate heuristics. Another approach is to transfer heuristics from **related problems**, which is particularly useful for entirely new problems. The LLM decomposes the new problem into components and matches these components with those of classic CO problems.

If a match is found, heuristics from the original problems can be transferred into new problem. Appendix A provides examples of the three generation methods.

When implementing the code, we provide **common reminders**, including input/output data formats, required libraries, annotations, and edge case considerations etc. to improve the quality of code. To reduce common errors, we optionally conduct a **smoke test**, where the LLM predicts the heuristic's output based on the detailed design and we then run the generated heuristic function. If the results are inconsistent or the code crashes, the error message is fed back to the LLM for adjustments until correct. For example, in the TSP, if the LLM expects a heuristic to select node A next but the heuristic either crashes or selects another node, the test fails and requires correction.

For novel problems without any reference, our approach supports to create basic algorithms like random ones and evolve them using methods from Section 3.1.2. The detailed workflow and prompts for the heuristic generation agent are provided in Appendix G.1.

### 3.1.2 HEURISTIC EVOLUTION AGENT

Relying solely on LLMs for heuristic evolution encounters inherent limitations due to constrained exploration capabilities and a lack of intrinsic motivation for evolution. Therefore, we employ a data-driven approach to enhance exploration capabilities in heuristic evolution.

**Single-round Evolution**    We adopt a data-driven heuristic evolution approach. Initially, we **run heuristic** on the training dataset to generate a baseline solution. Subsequently, we iteratively **perturb the original solution**, seeking enhancements or discontinuing if no progress is evident. The LLM then compares the two solutions and **identifies bottlenecks** that could affect the quality of the solution. For each identified bottleneck, we **reproduce the scenario** leading up to it independently, the LLM **proposes a suggestion** to navigate past the bottleneck, and we implement the recommendation to **verify the suggestion**. Should the solution quality improve, the LLM **summarizes the experience** from this instance and assimilates the effective recommendation. Ultimately, the LLM **updates the heuristic** with the validated recommendations. Figure3 illustrates this evolutionary process using the nearest neighbor heuristic as an exemplar within the TSP context. The comprehensive workflow and prompts for the single evolutionary round are detailed in AppendixG.2.

**Multi-round Evolution**    For further evolution, multi-round evolution is essential. Different data may yield various heuristics; thus additional validation data is required to filter effective heuristics for subsequent rounds. Both execution performance and execution time must be considered. Figure 4 displays the performance of multiple rounds of evolution for the nearest neighbor in the TSP.

### 3.2 PROBLEM SOLVING PHASE

As shown in Figure 5, before solving the problem, the benchmark evaluation agent provides feature extractors, and the heuristic selection agent dynamically selects heuristics during the problem solving process based on various instances and states.

### 3.2.1 BENCHMARK EVALUATION AGENT

Handling data directly can be challenging for LLMs, necessitating key feature extraction to reduce data dimensionality for efficient processing (Achiam et al., 2023; Zawbaa et al., 2018). Surface-level features often fail to capture problem complexity, requiring deeper features that describe both instance data and current solutions (Guan et al., 2021; Kim & Lee, 2019). Therefore, we built the benchmark evaluation agent to generate instance and solution feature extractors, providing detailed features for heuristic selection, as shown in Figure 5.

These feature extractors concentrate on **distinct characteristics** to discern between various instances, **effective representation** to alleviate computational demands, **characteristic attributes** for distinguishing between solution phases, **detailed insights** to pinpoint specific traits, and **comprehensive evaluations** to gauge the progress, quality, and scope of the solution. Table 5 in Appendix E details the features generated by the agent for different CO problems. The detailed workflow and prompts for the evaluation benchmark agent are provided in Appendix G.3.

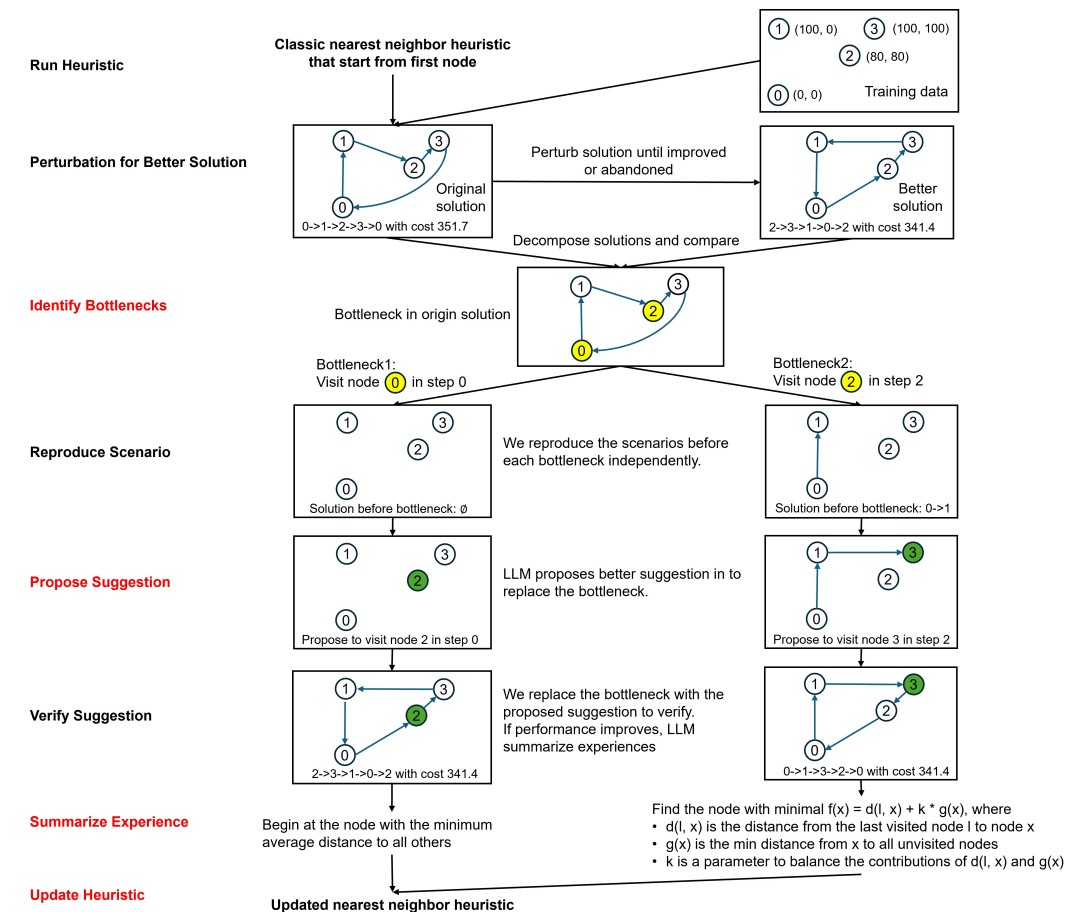

Figure 3: Single-round evolution for the nearest neighbor heuristic in TSP. The red text indicates interactions with the LLM. Evolution Round 1 in Appendix B. indicates the evolved code.

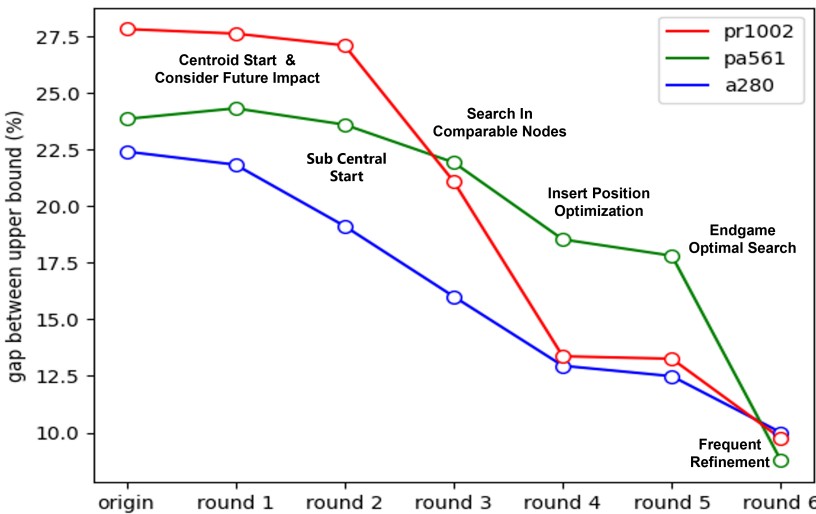

Figure 4: Performance of multi-round evolution on the nearest neighbor heuristic for TSP on pr1002, pcb561, a280 from TSPLIB. A smaller gap indicates better performance. The detailed evolved codes can be found in Appendix B.

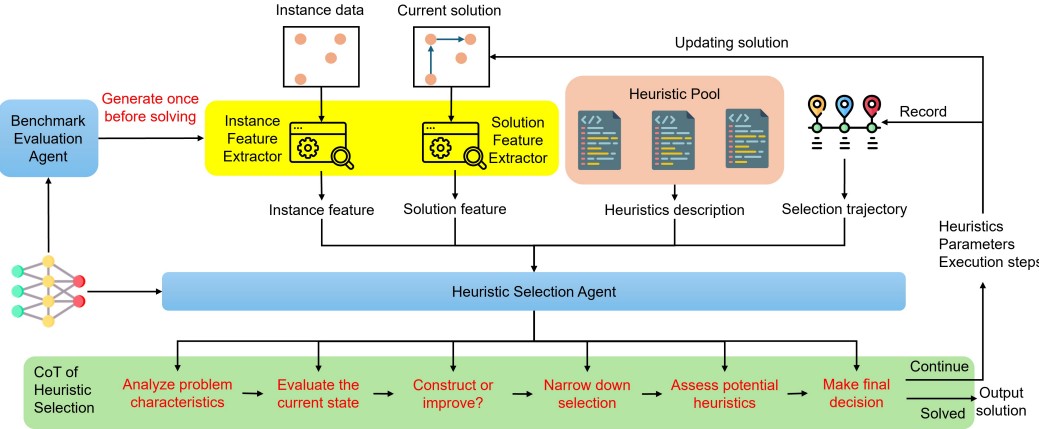

Figure 5: The problem solving process. The red text indicates interactions with the LLM. The Chain of Thought (CoT) for heuristic selection is completed in one query.

### 3.2.2 HEURISTIC SELECTION AGENT

The performance of heuristics is significantly influenced by the diversity of instances, making it crucial to dynamically select the most appropriate heuristic based on varying data characteristics (Burke et al., 2006). Different stages of the problem solving process also require distinct heuristics for effective optimization (Guan et al., 2021). Therefore, we dynamically select different heuristics for various instances and stages of problem solving.

As shown in Figure 5, for each round of selection, the heuristic selection agent receives information including instance features, solution features, descriptions of available heuristics, and selection trajectory, then makes the decision of the heuristic, parameters, and execution steps. The decision-making process is completed in one query with the following steps: **analyze problem characteristics** based on instance features such as scale and distribution, **evaluate the current state** to determine the progress and phase of the current solution using solution features, determine whether to **construct or improve** the solution based on both instance and solution features, **narrow down the selection** of suitable heuristics based on their descriptions, **assess potential heuristics** with the selection trajectory, and then **make final decision**.

Appendix F summarizes common selection patterns observed in LLMs without human guidance. The detailed workflow and prompts for the heuristic selection agent are provided in Appendix G.4.

## 4 EXPERIMENTS

In this section, we conducted experiments on HeurAgenix using GPT-4 as the foundational LLM. We assessed the complete workflow, including heuristic generation, evolution, benchmark evaluation, and selection, for both classic CO problems (Section 4.1) and new CO problems (Section 4.2), compared our evolution approach with state-of-the-art methods (Section 4.3) and combined our work with other hyper-heuristics (Section 4.4). For the detailed setting for whole experiment and dataset, please refer to Appendix D.

### 4.1 EXPERIMENTS ON CLASSIC PROBLEMS

We conducted experiments on five classic CO problems: the Traveling Salesman Problem (TSP), Capacitated Vehicle Routing Problem (CVRP), Job Shop Scheduling Problem (JSSP), Maximum Cut Problem (MaxCut), and Multidimensional Knapsack Problem (MKP). For problem details, refer to Appendix H.

To validate performance, we use the average gap defined by $\text{average\_gap} = \frac{1}{n} \sum_{i=1}^{n} \left| \frac{v_i - v_i^u}{v_i^u} \right| \times 100\%$, where $n$ is the number of test instances, $v_i$ is the heuristic value for the $i$-th test instance (e.g. tour

cost in TSP) and $v_i^u$ is the corresponding best known or upper bound. Variance is assessed using the average standard error of the mean (SEM) as average_sem $= \frac{1}{n} \sum_{i=1}^{n} \frac{\sigma_i}{\sqrt{m_i}}$, where $n$ is the number of test instances, $m_i$ is the experiment times on the $i$-th test instance, and $\sigma_i$ denotes the standard error on the $i$-th test instance. A lower gap indicates better performance, and a lower sem suggests less variance. These settings are used throughout the rest of the paper unless otherwise specified.

**Heuristic Generation and Evolution Experiment**   We conducted experiments on five classic problems to test the basic heuristics generated by the heuristic generation agent and the evolved heuristics from the heuristic evolution agent. Each experiment contains seven instances from publicly available academic datasets.

Figure 6 summarizes the experimental results, and the full experimental results and analyses are

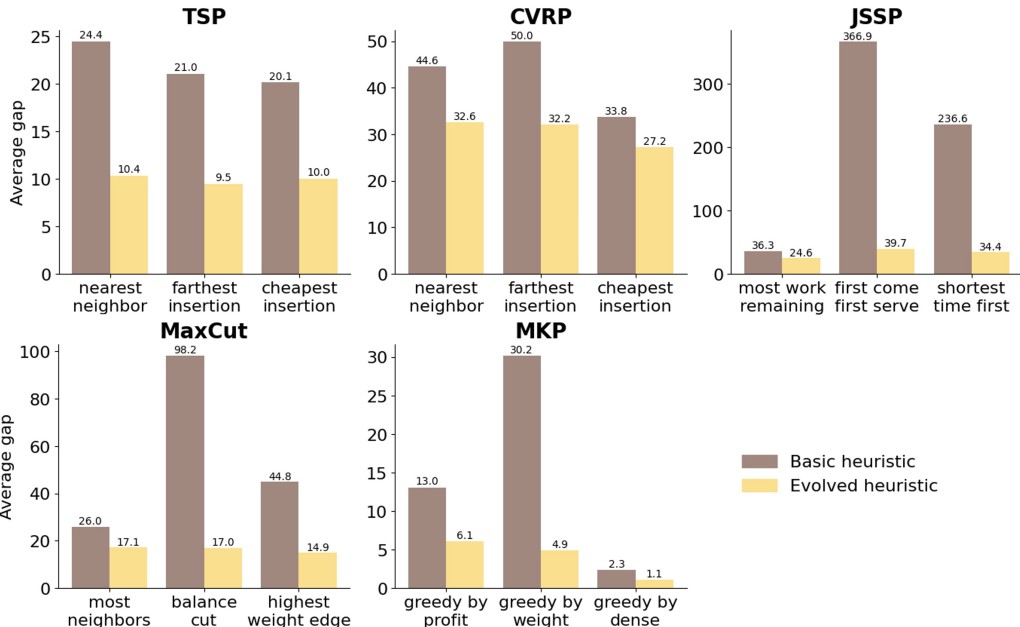

Figure 6: Heuristic generation and evolution experiment results. For each problem, we evolved three basic deterministic heuristics and compared their average gap.

provided in Table 6 in Appendix E. The experiments demonstrate that our HeurAgenix can correctly generate heuristic algorithms and effectively evolve them across different problems, even the basic heuristic's performance is poor, such as "first come first serve" in JSSP and "balance cut" in MaxCut.

**Heuristic Selection Experiment**   We evaluated the heuristic selection agent using both basic and evolved heuristic pools on the same test instances and employed random selection from corresponding heuristic pools as our baseline.

Figure 7 summarizes the experimental results, and the full experimental results and analyses are provided in Table 7 in Appendix E. These results show that the heuristic selection agent yields better performance with lower fluctuation than random selection. Additionally, selecting heuristics from the evolved heuristics pool yields better performance compared to selecting from the basic heuristics pool. Combining the results from Figure 6 and Figure 7, it is shown that the dynamic selection heuristic is better than single heuristics, indicating that heuristic selection agent works well.

### 4.2    Experiments on a New Problem

In this section, we introduce a novel, real-world, production-related, and complex CO problem: the Dynamic Production Order Scheduling Problem (DPOSP) to validate the effectiveness of HeurAgenix for new CO problems. DPOSP involves multiple production lines producing various products with

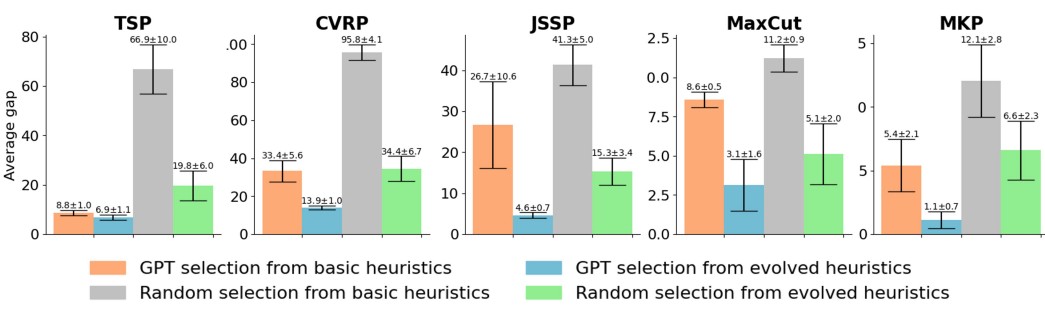

Figure 7: Results of heuristic selection experiments. Each experiment was conducted multiple times to reduce fluctuations, and the error bars (I-bars) represent the average sem.

transition times between products. Each order specifies the required product, quantity, and deadline, and all orders share the same priority. The objective is to fulfill as many orders as possible before their respective deadlines. For a detailed introduction, please refer to Appendix C.

Addressing novel problems, LLMs frequently face challenges in devising suitable heuristic algorithms. In DPOSP, even in the absence of order prioritization and production line capacity constraints within DPOSP, GPT-4 may nonetheless generate non-executable heuristics influenced by these hallucinated characteristics. To mitigate this, we adopt the heuristic transfer method mentioned in Section 3.1.1 to generate heuristics. Through this method, we have demonstrated that GPT-4 is capable of adeptly mapping the `vehicle`, `node`, `demands`, `travel_time` and `service_time` components in CVRP to the analogous `production_line`, `order`, `order_quantity`, `transition_time` and `production_time` in DPOSP. For detailed subsequent transferred code, we refer interested readers to Appendix A.3.

The test data and results in Table 2 show HeurAgenix works well on transfer heuristics from related problems, heuristic evolution, and heuristic selection for new CO problem.

Table 2: DPOSP experimental results. Heuristics marked with (*) are evolved versions. Solver results represent upper bounds ("-" indicates incomplete within one hour). The lower bound is provided by a random algorithm (not random heuristic selection). Higher fulfilled order numbers indicate better performance. The best results are in **bold**, and the second-best results are underlined.

| | | | Data | | | | |
|---|---|---|---|---|---|---|---|
| production line num | 5 | 5 | 5 | 10 | 10 | 20 | 20 |
| product num | 5 | 10 | 10 | 20 | 20 | 40 | 40 |
| order num | 10 | 50 | 100 | 100 | 200 | 500 | 2000 |
| order deadline | 12h | [0h, 24h] | [0h, 48h] | [0h, 24h] | [0h, 48h] | [0h, 120h] | [0h, 480h] |
| | | | Fulfilled Order Num | | | | |
| shortest operation | 8 | 40 | 76 | 43 | 138 | 344 | 1416 |
| shortest operation(*) | **10** | 40 | 82 | 46 | 144 | 378 | 1451 |
| least order remaining | 5 | 40 | 62 | 37 | 118 | 300 | 1130 |
| least order remaining(*) | 9 | 39 | 66 | 40 | 140 | 371 | 1386 |
| greedy by order density | 9 | 43 | 69 | 45 | 118 | 328 | 1388 |
| greedy by order density(*) | **10** | 44 | 82 | 51 | 130 | 392 | 1420 |
| LLM selection (basic) | 9.7±0.2 | 44.0±0.0 | 77.7±2.4 | 46.5±0.3 | 134.8±2.2 | 358.0±1.2 | 1482.7±3.0 |
| LLM selection (evolved) | **10.0±0.0** | 44.7±0.2 | 82.2±0.4 | 50.0±0.4 | 143.6±1.5 | **395.0±3.0** | **1492.8±1.3** |
| random selection (basic) | 8.8±0.4 | 38.4±1.3 | 66.2±1.3 | 41.8±0.3 | 120.4±0.8 | 325.2±1.9 | 1198.0±6.7 |
| random selection (evolved) | 9.6±0.4 | 42.3±0.9 | 72.3±1.2 | 47.2±1.1 | 132.8±3.0 | 344.5±1.8 | 1398.7±3.4 |
| random(lower bound) | 7.8±0.5 | 31.3±0.7 | 31.3±1.0 | 31.7±2.2 | 71.7±5.4 | 110.67±5.5 | 381.0±18.4 |
| results from solver(upper bound) | **10** | **46** | **85** | **52** | **152** | - | - |

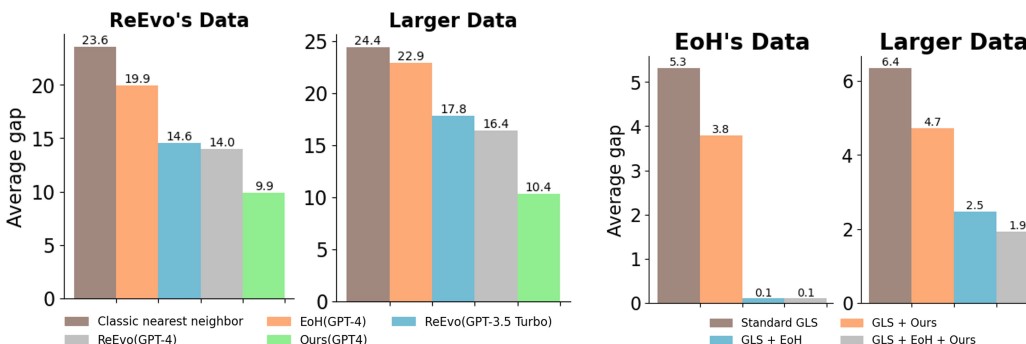

Figure 8: Evolution comparison results.    Figure 9: Combination results with GLS.

## 4.3 COMPARISON WITH OTHER EVOLUTION ALGORITHMS

We conduct a comparison of our heuristic evolution method against the approaches presented in EoH (Liu et al., 2024a) and ReEvo (Ye et al., 2024), using the nearest neighbor heuristic for TSP as a common benchmark. To ensure a fair comparison, we reran all EoH and ReEvo on GPT-4. and result of ReEvo (GPT-3.5 Turbo) is sourced from ReEvo's paper.
The experiments were conducted on both the test instances used in ReEvo's paper and another selected instances with a larger number of nodes. Figure 8 summarizes the experimental results, and the full experimental results and analyses are provided in Table 8 in Appendix E. These results indicate that our heuristic evolution method surpasses existing evolution algorithms based on LLMs.

## 4.4 COMBINATION WITH OTHER HYPER-HEURISTICS

We further explore the potential of HeurAgenix within hyper-heuristic frameworks. In this section, we aim to enhance the performance of Guided Local Search (GLS) (Voudouris & Tsang, 1999) by generating initial solutions using our evolved heuristic. We conducted four sets of experiments: (1) GLS with the classic nearest neighbor heuristic (**GLS**), (2) GLS with our evolved nearest neighbor heuristic (**GLS + Ours**), (3) GLS with the classic nearest neighbor heuristic and the updated distance matrix from EoH (**GLS + EoH**), and (4) GLS with our evolved nearest neighbor heuristic and the updated distance matrix from EoH (**GLS + EoH + Ours**).
The experiments were conducted on both the test instances used in EoH's paper and another selected instances with a larger number of nodes. Figure 9 summarizes the experimental results, and the full experimental results and analyses are provided in Table 9 in Appendix E. These results indicate that HeurAgenix can significantly enhance the capabilities of GLS.

## 5 CONCLUSION AND FUTURE WORK

We propose a multi-agent LLM-based paradigm, HeurAgenix, that employs LLMs to generate, evolve, evaluate, and select heuristic strategies for addressing CO problems. Our framework can effectively generate diverse heuristics for both classic and novel CO problems, showcasing its remarkable **adaptability and flexibility**. The **data-driven** evolution process enables the efficient evolution of heuristics without the need for prior knowledge, while the dynamically heuristic selection ensures **robustness** by continuously adapting to specific problem instance and the current state.

In the future, we will improve the efficiency of the generated code by enhancing the quality of heuristic code through supervised fine-tuning of open-source LLMs (Poesia et al., 2022). Additionally, we will enable LLMs to analyze larger instance data during the evolution phase by integrating data mining technique (Fink et al., 2023; Wan et al., 2024). We aim to improve the rationality of heuristic selection in the selection phase by exploring multiple LLM-enhanced machine learning algorithms, such as LLM-enhanced decision trees (Li et al., 2023), LLM-enhanced unsupervised learning techniques (Jung et al., 2024), and LLM-enhanced reinforcement learning approaches (Kwon et al., 2023; Liu et al., 2024b).

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

# A HEURISTIC GENERATION EXAMPLE

## A.1 GENERATE FROM LLMs INTERNAL KNOWLEDGE EXAMPLE

The following code is the original nearest neighbor heuristic for TSP, generated from LLMs' internal knowledge. The heuristic generation agent generates complete code with annotations, and here, for brevity, some content is omitted.

> **Nearest Neighbor In TSP**
>
> ```python
> def nearest_neighbor_f91d(
>     global_data: dict,
>     state_data: dict,
>     algorithm_data: dict,
>     get_state_data_function: callable
> ) > tuple[AppendOperator, dict]:
>     """Implements the nearest neighbor heuristic for the TSP problem.
>     Starting from an first city, at each step extend the tour by moving from the current city to
> its nearest unvisited neighbor.
>     Args:...
>     Returns:...
>     """
>     # Retrieve necessary data from global_data and state_data
>     ...
>
>     # If the tour is empty, start from first node.
>     if not current_solution.tour:
>         start_node = unvisited_nodes[0]
>         return AppendOperator(start_node), {}
>
>     # If all nodes are visited, return an empty operator
>     if no unvisited_nodes:
>         return None, {}
>
>     min_cost = float('inf')
>     # Find the nearest unvisited node to the last visited node
>     for node in unvisited_nodes:
>         cost = distance_matrix[last_visited][node]
>         if cost < min_cost:
>             nearest_node = node
>             min_cost = cost
>
>     # Insert the nearest at the end of the current solution
>     return AppendOperator(node=nearest_node), {}
> ```

Some additional remarks:

- The function name ends with a unique 4-digit identifier (`f91d` in this example) to avoid naming conflicts.

- The input consists of `instance_data`, `state_data`, and `algorithm_data`, which store instance data, current state data, and control parameters, respectively. The `get_state_data_function` receives a new solution and returns its state dictionary, useful for validating operations in complex problems, though not used here.

- The output consists of the current solution's operation and additional information. In this example, `AppendOperator(node)` adds a node to the end of the current tour. Other TSP heuristics may use `InsertOperator`, `SwapOperator`, `ReverseSegmentOperator`, etc. Some algorithms may output additional informa-

702
703
704
705
706
707
708
709
710
711
712
713
714
715
716
717
718
719
720
721
722
723
724
725
726
727
728
729
730
731
732
733
734
735
736
737
738
739
740
741
742
743
744
745
746
747
748
749
750
751
752
753
754
755

tion for controlling subsequent processes (e.g., updated pheromone levels in ant colony algorithms), but these are empty here.

## A.2   Learn from Reference Paper Example

The following heuristic is generated from "Fast minimum-weight double-tree shortcutting for Metric TSP: Is the best one good enough?" (Deineko & Tiskin, 2010) available at `https://arxiv.org/abs/0710.0318`. Below is the main work summarized and judged by GPT-4:

> Based on the content provided from the "Heuristic improvements" section, the paper discusses two heuristic improvements for the minimum-weight double-tree shortcutting algorithm: one aimed at increasing the speed and the other at improving the approximation quality.
>
> The first heuristic involves limiting the search across a subtree to a depth-$k$ list of a node for a chosen value of $k$, which dramatically improves running time without significantly affecting approximation quality. The second heuristic expands the search space by constructing a new tree that allows an enlarged tour neighborhood, increasing the likelihood of finding a better solution.
>
> These heuristic improvements are substantive and can be implemented in your framework. The bounded neighbor lists and the degree-increasing operation can be incorporated into the minimum-weight double-tree shortcutting algorithm to form a new heuristic.
>
> Given that we have detailed information on the algorithm and the heuristic improvements, we appear to have enough to implement a heuristic for the TSP problem based on this paper. We can name the heuristic based on the improvements outlined, indicating the use of bounded neighbor lists and degree-increasing operations.

These are the heuristic descriptions from GPT-4:

> Let G be a complete graph representing the TSP problem on n nodes, with a weight function w: $E(G) \to R^+$ representing the distances between nodes. Let T be a minimum spanning tree (MST) of G rooted at a node of degree 1. The heuristic operates on T to generate a Hamiltonian cycle H approximating the optimal TSP tour.
>
> 1. Depth-k limitation: Define a tree distance $td(u, v)$ as the number of edges in the unique path between nodes u and v in T. For a fixed parameter k, the heuristic considers only nodes v in T(u) such that $td(u, v) \leq k$ when searching for minimum-weight paths during the upsweep and downsweep stages of the algorithm.
>
> 2. Degree-increasing operation: For a node u in T with a child v that is not a leaf and for which $deg(u) + deg(v) \leq D$ (a fixed parameter), redefine all children of v to be children of u, effectively making v a leaf and expanding the search space for tours.
>
> The heuristic then constructs a Hamiltonian cycle H by shortcutting an Eulerian tour of the doubled tree T' obtained from T after applying the degree-increasing operations. The shortcutting process ensures that for each node u in T, all nodes of the subtree rooted at u appear consecutively in H.

These are generated code, here we only show the main heuristic function, which will generate the complete code after actual execution:

**Double Tree Heuristic With Improvements In TSP**

```python
def double_tree_heuristic_with_improvements_d1fd(
    global_data: dict,
    state_data: dict,
    algorithm_data: dict,
    get_state_data_function: callable,
    depth_k=2,
    depth_D=5
) -> tuple[InsertOperator, dict]:
    """Double Tree Heuristic with Improvements heuristic algorithm for the TSP problem.
    This heuristic is based on the minimum-weight double-tree shortcutting algorithm, with
    enhancements for limited search depth and expanded tour neighborhoods.
    Args:
        ...
        depth_k (int): The depth limit for the limited search across subtrees (default: 2).
        degree_D (int): The degree limit for the expanded tour neighborhood (default: 5).
    Returns: ...
    """
    # Extract necessary data from global_data and state_data
    ...

    # Step 1: Construct the Minimum Spanning Tree (MST)
    mst = construct_minimum_spanning_tree(global_data["distance_matrix"])

    # Step 2: Apply the degree-increasing operation to expand the tour neighborhood
    expanded_tree = apply_degree_increasing_operation(mst, degree_D)

    # Step 3: Double the edges of the expanded tree to create an Eulerian graph
    eulerian_graph = double_edges(expanded_tree)

    # Step 4: Find an Eulerian tour of the Eulerian graph
    euler_tour = find_eulerian_tour(eulerian_graph)

    # Step 5: Perform depth-limited search to find minimum-weight shortcutting of
    the Euler tour
    shortcutting_path = depth_limited_search(euler_tour, depth_k)

    # Step 6: Convert the shortcutting path to a Hamiltonian cycle (tour) and update
    the solution
    hamiltonian_cycle = convert_to_hamiltonian_cycle(shortcutting_path)

    # Verify the new solution is valid before returning the operator
    if not get_state_data_function(hamiltonian_cycle)['is_valid']:
        return None, {} # If the solution is not valid, return None

    # Generate the operator that leads to the new solution
    operator = generate_operator_from_hamiltonian_cycle(hamiltonian_cycle,  cur-
rent_solution)
    return operator, {}
```

## A.3 TRANSFER FROM RELATED PROBLEM EXAMPLE

GPT-4 decomposes the CVPR and DPOSP, and maps the corresponding components, resulting in Table3.

Table 3: Component Mapping from CVRP to DPOSP

| CVRP Component | DPOSP Component |
|----------------|-----------------|
| vehicle | production_line |
| node | order |
| unvisited_nodes | unfulfilled_orders |
| demands | order_quantity |
| travel_time | transition_time |
| service_time | production_time |

Based on the mapping in Table3, the GPT-4 can obtain the transferred code as:

**Nearest Neighbor In CVRP**

```
def nearest_neighbor_99ba(
    global_data: dict, state_data: dict,
    state_data: dict,
    algorithm_data: dict,
    get_state_data_function: callable
) -> tuple[AppendOperator, dict]:
    """Nearest Neighbor heuristic for CVRP.
    This algorithm iterates over each vehicle, starting from the depot. For each vehicle, it
    finds the nearest unvisited node and appends it to the route, respecting the vehicle's capacity
    constraints.
    This process repeats until no further nodes can be visited without exceeding the vehicle's
    capacity or all nodes have been visited.
    Args:...
    Returns:...
    """
    # Retrieve necessary data from global_data and state_data
    ...
    # Iterate over each vehicle
    for vehicle_id, remaining_capacity in enumerate(remaining_vehicle):
        if not unvisited_nodes or remaining_capacity <= 0:
            # If there are no unvisited nodes or the vehicle has no remaining capacity, continue to
            the next vehicle
            continue

        last_visited    =    depot    if    not    current_solution[vehicle_id]    else
        current_solution[vehicle_id][-1]
        nearest_node = None
        min_distance = float('inf')
        # Find the nearest unvisited node that doesn't exceed the vehicle's capacity
        for node in unvisited_nodes:
            if demands[node] <= remaining_capacity
                and distance_matrix[last_visited][node] < min_distance:
                nearest_node = node
                min_distance = distance_matrix[last_visited][node]
        if nearest_node is not None:
            position=len(current_solution[vehicle_id])
            return InsertOperator(vehicle_id, nearest_node, position), {}
    # If no vehicle has no remaining capacity, return None
    return None, {}
```

**Shortest Operation In DPOSP**

```python
def shortest_operation_ff40(
    global_data: dict,
    state_data: dict,
    algorithm_data: dict,
    get_state_data_function: callable
) -> tuple[AppendOperator, dict]:
    """Shortest Operation Heuristic for DPOSP.
    This heuristic iterates over each production line and attempts to insert the shortest unfulfilled order
    that can be appended to the end of the production line's schedule without violating any constraints.
    Args:...
    Returns:...
    """
    # Retrieve necessary data from global_data and state_data
    ...

    # Check if there are any feasible orders to fulfill
    if not feasible_orders_to_fulfill:
        return None, {}

    # Sort unfulfilled orders based on their quantity (shortest first)
    sorted_orders = sorted(feasible_orders_to_fulfill,
        key=lambda order: order_quantity[order])

    # Iterate over each production line
    for line_id in range(production_line_num):
        # Iterate over the sorted unfulfilled orders
        for order_id in sorted_orders:
            # Check if the production line can produce the product
            product_id = global_data["order_product"][order_id]
            if production_rate[line_id][product_id] == 0:
                continue
            # Generate a new schedule by appending the order to the selected production line
            new_schedule = current_solution.production_schedule[line_id][:]
            new_schedule.append(order_id)
            # Validate the new schedule
            if validation_single_production_schedule(line_id, new_schedule):
                # If valid, create and return the InsertOperator
                position=len(new_schedule) - 1
                return InsertOperator(line_id, order_id, position), {}
    # If no valid operation is found, return None
    return None, {}
```

## B  HEURISTIC EVOLUTION EXAMPLE

The following evolution codes show the evolution process for the nearest neighbor in TSP. The ~~red text~~ indicates deleted content, and the green text indicates added content.

**Evolution Round 1: Centroid Start And Consider Future Impact**

```
...
# If the tour is empty, start from node with the lowest average distance to all other nodes
if not current_solution.tour:
    start_node = unvisited_nodes[0]
    avg_distances = [np.mean([
        distance_matrix[i][j] for j in range(node_num)])
        for i in range(node_num)]
    start_node = np.argmin(avg_distances)
    return AppendOperator(start_node), {}
...
# Utilize f(x) = d(l, x) + k * g(x) to weigh immediate and future node distances
future_ratio = algorithm_data.get("future_ratio", 0.20)
for node in unvisited_nodes:
    min_distance = distance
    future_cost = np.min([
        distance_matrix[node][other]
        for other in unvisited_nodes if node != other])
    cost = distance_matrix[last_visited][node]
        + future_ratio * future_cost
    if distance < min_distance:
        nearest_node = node
        min_distance = distance
...
```

**Evolution Round 2: Sub-Central Nearest Start**

```
...
# If the tour is empty, start from node with the lowest average distance to all other nodes
if not current_solution.tour:
    avg_distances = [np.mean([
        distance_matrix[i][j] for j in range(node_num)
        ])for i in range(node_num)]
    start_node = np.argmin(avg_distances)
    start_node = np.argsort(avg_distances)[1]
    return AppendOperator(start_node), {}
...
```

**Evolution Round 3: Search In Comparable Nodes**

```
...
future_ratio = algorithm_data.get("future_ratio", 0.20)
significance_threshold = algorithm_data.get("significance_threshold", 0.30)
comparable_threshold = algorithm_data.get("comparable_threshold", 1.20)
nearest_node = min(unvisited_nodes,
    key=lambda node: distance_matrix[last_visited][node])
nearest_distance = distance_matrix[last_visited][nearest_node]

# If distance of nearest neighbor is significantly shorter than others, insert the
nearest neighbor
avg_distance = np.mean([
    distance_matrix[last_visited][node] for node in unvisited_nodes])
if nearest_distance < significance_threshold * avg_distance:
    return AppendOperator(node), {}

# Evaluate multiple unvisited nodes with comparable distances
comparable_distance = comparable_threshold * nearest_distance
comparable_nodes = [node for node in unvisited_nodes
    if distance_matrix[last_visited][node] <= comparable_distance]
for node in unvisited_nodes:
for node in comparable_nodes:
    future_cost = np.min([
        ...
```

**Evolution Round 4: Insert Position Optimization**

```
...
best_increase = float('inf')
for node in comparable_nodes:
    future_cost = np.min([
        distance_matrix[node][other]
        for other in unvisited_nodes if node != other])
    cost = distance_matrix [last_visited][node]
        + future_ratio * future_cost
    if distance < min_distance:
        nearest_node = node
        min_distance = distance
    for i in range(len(current_solution.tour) + 1):
        if i == 0:
            next_node = current_solution.tour[0]
            cost_increase = distance_matrix[node][next_node]
        elif i == len(current_solution.tour):
            prev_node = current_solution.tour[-1]
            cost_increase = distance_matrix[prev_node][node]
        else:
            prev_node = current_solution.tour[i - 1]
            next_node = current_solution.tour[i]
            cost_increase = \
                distance_matrix[prev_node][node] \
                + distance_matrix[node][next_node] \
                - distance_matrix[prev_node][next_node]
        if cost_increase < best_increase:
            best_increase, best_node, best_position = cost_increase, node, i
    return InsertOperator(node=best_node, position=best_position), {}
```

**Evolution Round 5: Endgame Optimal Search**

```
    ...
    future_ratio = kwargs.get("future_ratio", 0.20)
    significance_threshold = kwargs.get("significance_threshold", 0.30)
    comparable_threshold = kwargs.get("comparable_threshold", 1.20)
    endgame_threshold = algorithmdata.get("endgame_threshold", 10)

    # If the number of unvisited nodes is less than the threshold, perform exhaustive
search
    if len(unvisited_nodes) < endgame_threshold:
        min_distance = float('inf')
        for perm in permutations(unvisited_nodes):
            # Calculate the distance for the rest path, including: the distance between last visited
node and rest path's start node, the total distance of rest path, and the distance between rest
path's end node and whole path's start node
            path_distance =
                distance_matrix[last_visited, perm[0]] \
                + sum(distance_matrix[perm[i], perm[i+1]] \
                    for i in range(len(perm) - 1)) \
                + distance_matrix[perm[-1], current_solution.tour[0]]
            # Update the shortest path
            if path_distance < min_distance:
                min_distance, best_path = path_distance, perm
        return AppendOperator(best_path[0]), {}
    best_increase = float('inf')
    for node in comparable_nodes:
        ...
```

**Evolution Round 6: Frequent Refinement**

```
future_ratio = kwargs.get("future_ratio", 0.20)
significance_threshold = kwargs.get("significance_threshold", 0.30)
comparable_threshold = kwargs.get("comparable_threshold", 1.20)
end_game_threshold = kwargs.get("end_game_threshold", 10)
apply_2opt_frequency = kwargs.get("apply_2opt_frequency", 5)
# Apply the 2-opt heuristic periodically
N = len(current_solution.tour)
if N > 2 and N % apply_2opt_frequency == 0:
    best_delta = 0
    best_pair = None

    for i in range(N - 1):
        for j in range(i + 2, N):
            if j == N - 1 and i == 0:
                continue

            a = current_solution.tour[i]
            b = current_solution.tour[(i + 1) % N]
            c = current_solution.tour[j]
            d = current_solution.tour[(j + 1) % N]
            current_cost = distance_matrix[a][b] + distance_matrix[c][d]
            new_cost = distance_matrix[a][c] + distance_matrix[b][d]
            delta = new_cost - current_cost

            if delta < best_delta:
                best_delta = delta
                best_pair = (i + 1, j)

    if best_pair:
        return ReverseSegmentOperator([best_pair]), {}
# If the number of unvisited nodes is less than the threshold, perform exhaustive search
if len(unvisited_nodes) < end_game_threshold:
    ...
```

# C   INTRODUCTION TO DPOSP

DPOSP involves multiple production lines, each capable of producing various products at different production speeds. When switching between different products on the production line, transition times are required, and no production occurs during these transitions. Each order specifies one required product, quantity, and deadline. Each order must be produced in its entirety on a single production line and completed before the deadline. Our objective is to maximize the number of completed orders, with each order having the same priority regardless of the quantity required.

To formally describe DPOSP, we build the following optimization model:

$$\text{Maximize} \quad \sum_i \sum_j I(X_{ij} \neq 0) \tag{1}$$

$$\text{subject to} \quad \sum_i \sum_j I(X_{ij} = k) \leq 1 \quad \forall k \tag{2}$$

$$s_{ij} = \begin{cases} 0 & \text{if } j = 1 \\ e_{i,j-1} + t_{i,P_{k_{j-1}},P_{k_j}} & \text{if } j > 1 \end{cases} \tag{3}$$

$$e_{ij} = s_{ij} + \frac{Q_k}{v_{iP_k}} \quad \text{if } X_{ij} = k \tag{4}$$

$$e_{ij} \leq D_k \quad \text{if } X_{ij} = k \tag{5}$$

where:

- $X_{ij}$ (Decision Variable): represents the $j$-th production action on the $i$-th production line, where $X_{ij} \in \{0, 1, \ldots, k\}$, with $X_{ij} = k$ indicating production of order $k$ and $X_{ij} = 0$ indicating no production action.
- $v_{ip}$ (Input Variable): production speed of production line $i$ for product $p$.
- $t_{i,p,p'}$ (Input Variable): transition time for production line $i$ from product $p$ to product $p'$.
- $Q_k$ (Input Variable): quantity required for order $k$.
- $P_k$ (Input Variable): product required for order $k$.
- $D_k$ (Input Variable): deadline for order $k$.
- $s_{ij}$ (Intermediate Variable): start time of the $j$-th production action on production line $i$.
- $e_{ij}$ (Intermediate Variable): end time of the $j$-th production action on production line $i$.
- $I(\cdot)$ (Indicator Function): equals 1 if the condition is true, and 0 otherwise.

## D  EXPERIMENT SETTINGS

Table 4: Detailed Parameters and Settings

| Feild | Item | Value |
|---|---|---|
| LLM Setting | | GPT-4, version 2024-05-01-preview, temperature 0.7, top-p 0.95, max tokens 1600 |
| TSP | Data Source | `http://comopt.ifi.uni-heidelberg.de/software/TSPLIB95/tsp/` |
| | Test Data | tsp225, a280, pcb442, pa561, gr666, pr1002, pr2392 |
| | Validation Data | brg180, eil101, gr202, pr124, pr152, rd100, u159 |
| | Training Data | 20 cases that sampled from other instances |
| | Generated(Evolved, Selected) Heuristics | ant colony, cheapest insertion, farthest insertion, greedy algorithm, greedy randomized adaptive search procedure grasp nearest insertion, nearest neighbor, random pairwise insertion, insertion heuristics, simulated annealing, 2opt, 3opt |
| CVRP | Data source | `http://vrp.galgos.inf.puc-rio.br/index.php/en/` |
| | Test Data | A-n80-k10, B-n78-k10, E-n101-k14, F-n135-k7, M-n200-k17, P-n101-k4, X-n1001-k43 |
| | Validation Data | A-n63-k10, B-n67-k10, E-n76-k10, F-n45-k4, M-n101-k10, P-n70-k10, X-n101-k25 |
| | Training Data | 20 cases that sampled from other instances |
| | Generated(Evolved, Selected) Heuristics | farthest insertion, greedy, min cost insertion, nearest neighbor node shift between routes, petal algorithm, saving algorithm, three opt, two opt |
| JSSP | Data source | `https://people.brunel.ac.uk/~mastjjb/jeb/orlib/files/` |
| | Test Data | LA05, LA10, LA15, LA20, LA25, LA30, LA35 |
| | Validation Data | LA01, LA06, LA11, LA16, LA21, LA26, LA31, LA36 |
| | Training Data | 20 cases that sampled from other instances |
| | Generated(Evolved, Selected) Heuristics | first come first served, least work remaining, longest job next, longest processing time first, most work remaining, shift operator, shortest job next, shortest processing time first, 2opt, 3opt |
| MaxCut | Data source | `https://grafo.etsii.urjc.es/optsicom/maxcut.html#instances` |
| | Test Data | g10, g20, g30, toursg3-15, toursg3-8, tourspm3-15-50, tourspm3-8-50 |
| | Validation Data | g1, g11, g21, g41, g51, sg3dl051000, sg3dl052000, sg3dl053000, sg3dl054000 |
| | Training Data | 20 cases that sampled from other instances |
| | Generated(Evolved, Selected) Heuristics | balanced cut, greedy swap, highest delta edge, highest delta node, highest weight edge, most weight neighbors, multi swap 2, simulated annealing |
| MKP | Data source | `https://people.brunel.ac.uk/~mastjjb/jeb/orlib/files/` |
| | Test Data | gmknap1 _1, mknap1 _7, mknapcb9-01, mknapcb9-11, mknapcb9-21, PB7.DAT, WEING1.DAT |
| | Validation Data | mknap1 _2, mknap1 _6, mknapcb9-02, mknapcb9-12, mknapcb9-22, PB1.DAT, SENTO1.DAT |
| | Training Data | 20 cases that sampled from other instances |
| | Generated(Evolved, Selected) Heuristics | block flip, greedy by cost benefit, greedy by density, greedy by least remaining capacity, greedy by profitto weight ratio, greedy by profit, greedy by resource balance, greedy by weight, greedy improvement, k flip, single swap heuristic, two opt |
| DPOSP | Data source | Sampled from distribution |
| | Generated(Evolved, Selected) Heuristics | exchange production orders, farthest deadline insertion, greedy by order density, greedy deadline proximity, greedy order selection, least order remaining, longest order next, maximum remaining work order, nearest order scheduling, order shift between lines, random, shortest operation, shortest order next, 2opt production sequence |
| Heuristic evolution setting | Max evolution round | 7 |
| | Running time limitation | within 3 times of the original heuristic |
| | Perturbation ratio | 0.1 |
| | Max perturbation times | 1000 |
| | Max filterd number for next round | 3 |
| Heuristic selection setting | Max steps | 2 times of task num (such as node num in TSP, order num in DPOSP) |
| | Max feature context length | 1000 |
| Queries Number for Evolution | EoH | Fixed strategies * population maximum iterations 5 * 10 * 20 = 1000 in our experiment for nearest neighbor |
| | ReEvo | related to population size and evolution 112 in our experiment for nearest neighbor |
| | HeurAgenix | varies based training samples, perturbation success rate, and bottleneck number 228 in our experiment for nearest neighbor |
| GLS setting | GLS searhing time | 10s |

## E  DETAILED EXPERIMENT RESULT

From the benchmark evaluation agent, we can get various features for both the instance and the solution. Despite the fluctuating outputs of the LLM, the core essential features can be extracted. Table 5 displays the common features of classic CO problems.

Table 6 shows the average gap of base heuristics (without *) from the heuristic generation agent and evolved heuristics (with *) from the heuristic evolution agent.

From Table 6, we can observe the following points:

- The same heuristic can perform differently under different data distributions. For example, the "farthest insertion" heuristic for the CVRP problem performs particularly well on datasets B-n78-k10, E-n101-k14, and F-n135-k7, but not on others. This verifies the statement that the performance of heuristics is significantly influenced by the diversity of problem data in Section 3.2.2.

- Most heuristics show significant improvement after evolution. For instance, in the TSP problem, the evolved "nearest neighbor" heuristic consistently outperforms the base heuristic across all datasets.

Table 5: Features from benchmark evaluation agent. Commonly considered features by the heuristic selection agent are in **bold**.

|  | Instance data feature | Current solution feature |
|---|---|---|
| TSP | **node num**, **average distance**, std dev distance, edge length distribution | **visited num**, **current cost**, **last visited**, nearest neighbors for last visited, unvisited edge length distribution |
| CVRP | **task num**, **vehicle num**, **capacity**, **average demands**, **average distance**, edge length distribution | **finished tasks**, **current cost**, max vehicle loads, min vehicle loads, **average vehicle loads**, fulfilled demands, remaining demands |
| JSSP | **job operation sequence**, job operation time, **job num**, **machine num**, total processing times | **finished jobs**, **job operation index**, job last operation end times, machine last operation end times, current makespan |
| MaxCut | **total nodes**, **total edges**, average weights, min weights, max weights, positive weight num, negative weight num | **selected num**, **set a count**, **current cut value**, average weight for unselected node |
| MKP | **item num**, **resource num**, average profit, max profit, min profit, average weight, max weight, min weight, average capacity, max capacity, min capacity | **current profit**, **current weight**, remaining capacity, **selected num**, profit per remaining capacity |

Table 6: Detailed heuristic generation and evolution experiment result. Heuristics without an (*) are basic heuristics that generated by the heuristics generation agent and heuristics with (*) are evolved heuristics that evolved by the heuristic evolution agent.

| Problem | Heuristic | Data | | | | | | |
|---|---|---|---|---|---|---|---|---|
| | | tsp225 | a280 | pcb442 | pa561 | gr666 | pr1002 | pr2392 |
| tsp | nearest neighbor | 28.35 | 22.41 | 22.03 | 23.85 | 24.67 | 27.82 | 21.99 |
| | nearest neighbor(*) | 5.31 | 10.00 | 11.99 | 8.76 | 13.72 | 9.74 | 12.9 |
| | farthest insertion | 18.12 | 23.85 | 22.31 | 24.14 | 17.7 | 19.56 | 21.7 |
| | farthest insertion(*) | 10.41 | 5.00 | 7.83 | 9.55 | 9.86 | 11.92 | 11.86 |
| | cheapest insertion | 14.49 | 13.07 | 18.86 | 21.5 | 19.19 | 25.01 | 28.82 |
| | cheapest insertion(*) | 7.43 | 8.10 | 8.23 | 8.54 | 14.98 | 11.16 | 11.74 |
| | | A-n80-k10 | B-n78-k10 | E-n101-k14 | F-n135-k7 | M-n200-k17 | P-n101-k4 | X-n1001-k43 |
| cvrp | nearest neighbor | 33.26 | 43.98 | 55.39 | 54.22 | 56.00 | 49.93 | 19.54 |
| | nearest neighbor(*) | 25.63 | 37.26 | 47.27 | 34.89 | 41.84 | 29.83 | 11.50 |
| | farthest insertion | 29.57 | 36.94 | 85.10 | 23.84 | 104.00 | 30.10 | 40.16 |
| | farthest insertion(*) | 26.61 | 33.61 | 44.44 | 23.32 | 47.02 | 28.09 | 22.10 |
| | cheapest insertion | 20.60 | 42.92 | 39.93 | 41.82 | 36.00 | 38.03 | 17.12 |
| | cheapest insertion(*) | 17.73 | 37.85 | 30.61 | 28.93 | 32.65 | 32.75 | 9.90 |
| | | LA05 | LA10 | LA15 | LA20 | LA25 | LA30 | LA35 |
| JSSP | most work remaining | 12.31 | 27.77 | 25.19 | 55.65 | 53.02 | 43.47 | 36.55 |
| | most work remaining(*) | 0.00 | 41.13 | 29.01 | 20.93 | 25.93 | 18.01 | 37.05 |
| | first come first serve | 200.0 | 253.03 | 227.17 | 332.93 | 522.72 | 524.06 | 508.42 |
| | first come first serve(*) | 38.84 | 47.86 | 46.45 | 36.11 | 40.31 | 28.71 | 39.81 |
| | shortest processing time first | 141.82 | 127.77 | 161.56 | 235.81 | 381.99 | 290.77 | 316.15 |
| | shortest processing time first(*) | 16.32 | 25.17 | 20.65 | 29.08 | 37.43 | 29.44 | 83.06 |
| | | g10 | g20 | g30 | toursg3-8 | toursg3-15 | tourspm3-8-50 | tourspm3-15-50 |
| MaxCut | most weight neighbors | 25.93 | 28.80 | 29.89 | 24.22 | 22.46 | 25.99 | 24.43 |
| | most weight neighbors(*) | 16.80 | 18.07 | 19.01 | 18.21 | 14.38 | 17.62 | 15.52 |
| | highest weight edge | 59.58 | 45.06 | 56.10 | 44.18 | 36.57 | 36.56 | 35.83 |
| | highest weight edge(*) | 17.85 | 14.13 | 18.22 | 13.27 | 11.37 | 14.10 | 15.38 |
| | balance cut | 96.29 | 95.54 | 99.82 | 98.75 | 98.73 | 98.24 | 99.87 |
| | balance cut(*) | 14.24 | 18.07 | 14.49 | 13.32 | 13.18 | 21.59 | 24.43 |
| | | mknap1_1 | mknap1_7 | WEING1.DAT | PB7.DAT | mknapcb9-01 | mknapcb9-11 | mknapcb9-21 |
| MKP | greedy by profit | 36.84 | 16.17 | 3.97 | 15.36 | 12.55 | 3.94 | 2.52 |
| | greedy by profits(*) | 0.00 | 4.26 | 3.97 | 15.36 | 12.55 | 3.94 | 2.52 |
| | greedy by weight | 36.84 | 38.68 | 33.24 | 43.19 | 27.01 | 21.46 | 10.79 |
| | greedy by weight(*) | 0.00 | 0.00 | 9.24 | 6.99 | 7.27 | 7.36 | 3.13 |
| | greedy by dense | 0.00 | 4.26 | 1.42 | 1.26 | 5.27 | 2.49 | 1.50 |
| | greedy by dense(*) | 0.00 | 0.00 | 1.40 | 1.06 | 2.69 | 1.35 | 1.07 |

- The heuristic evolution agent effectively improves heuristics, even the origin heurisitic performance is poor. For example, the "first come first serve" heuristic for the JSSP problem and the "balance cut" heuristic for the MaxCut problem both show substantial improvements after evolution.
- Similar to machine learning algorithms, heuristic evolution effectiveness is influenced by training data. In some cases, "overfitting" may occur, leading to poor results on certain datasets. For instance, the "most work remaining" heuristic for the JSSP problem performs poorly on the LA10, LA15, LA35 dataset, indicating potential overfitting.

Table 7 shows the average gap of LLM selection from basic heuristics(LLM (B)), LLM selection from evolved heuristics (LLM (E)), random selection from basic heuristics (Random (B)) random selection from evolved heuristics(Random (E)).

Table 7: Detailed heuristic selection experiment result. Each experiment was conducted multiple times and the ± represent the standard errors of the mean (SEMs) for the results. The best results are highlighted in **bold**, and the second-best results are underlined.

| Problem | Function | Data | | | | | | |
|---|---|---|---|---|---|---|---|---|
| | | tsp225 | a280 | pcb442 | pa561 | gr666 | pr1002 | pr2392 |
| TSP | LLM (B) | 7.2±0.99 | 9.79±1.06 | 6.73±1.59 | 9.86±0.82 | 9.07±0.85 | 8.45±0.4 | 10.21±1.58 |
| | LLM (E) | **3.96±1.07** | **7.06±1.55** | **10.81±1.29** | **6.88±1.2** | **7.45±0.8** | **5.29±1.15** | **6.82±0.42** |
| | Random (B) | 48.61±3.48 | 63.55±12.73 | 65.03±9.62 | 63.92±6.85 | 90.28±11.5 | 98.48±19.97 | 38.37±5.5 |
| | Random (E) | 12.02±3.62 | 19.92±5.56 | 14.88±4.73 | 17.27±8.61 | 22.02±9.77 | 31.74±2.89 | 20.57±7.03 |
| | | A-n80-k10 | B-n78-k10 | E-n101-k14 | F-n135-k7 | M-n200-k17 | P-n101-k4 | X-n1001-k43 |
| CVRP | LLM (B) | 26.14±6.0 | 29.2±8.88 | 43.28±4.48 | 41.95±6.68 | 41.78±4.59 | 27.49±5.6 | 23.62±3.24 |
| | LLM (E) | **13.12±0.17** | **20.57±1.6** | **21.83±0.79** | **10.62±1.16** | **17.18±0.72** | **6.74±0.67** | **7.49±1.92** |
| | Random (B) | 58.73±3.04 | 72.6±4.54 | 79.74±6.29 | 105.63±2.31 | 128.6±5.28 | 94.39±5.21 | 130.69±2.27 |
| | Random (E) | 23.57±9.65 | 51.62±4.71 | 33.64±4.35 | 37.0±2.34 | 42.88±14.48 | 31.07±8.08 | 21.26±3.11 |
| | | LA05 | LA10 | LA15 | LA20 | LA25 | LA30 | LA35 |
| JSSP | LLM (B) | 21.92±18.36 | 10.68±5.92 | 22.78±7.56 | 34.24±11.15 | 40.57±12.36 | 38.45±15.65 | 18.49±2.91 |
| | LLM (E) | **0.00±0.00** | **0.00±0.00** | **6.17±0.53** | **6.18±2.44** | **6.86±0.26** | **10.17±0.78** | **12.8±0.87** |
| | Random (B) | 23.24±5.12 | 17.49±2.67 | 26.91±1.48 | 60.89±7.72 | 62.21±7.08 | 53.49±7.76 | 44.94±3.12 |
| | Random (E) | 12.2±2.2 | 10.2±4.2 | 9.09±4.44 | 34.19±3.53 | 18.83±1.87 | 12.14±2.51 | 10.74±4.73 |
| | | g10 | g20 | g30 | toursg3-8 | toursg3-15 | tourspm3-8-50 | tourspm3-15-50 |
| MaxCut | LLM (B) | 7.97±0.72 | 9.86±1.22 | 9.73±0.46 | 8.35±0.0 | 6.65±0.21 | 9.14±0.91 | 8.3±0.0 |
| | LLM (E) | **1.85±1.69** | **2.59±1.91** | **3.84±0.88** | **2.45±0.86** | 3.5±2.02 | **3.55±2.66** | **4.2±1.43** |
| | Random (B) | 12.34±1.09 | 10.39±0.64 | 12.35±0.66 | 11.79±0.78 | 8.35±1.15 | 13.04±0.83 | 10.27±1.01 |
| | Random (E) | 4.63±1.44 | 8.73±2.4 | 7.06±2.08 | 6.3±2.11 | 8.2±1.26 | 10.25±1.67 | 6.7±1.71 |
| | | mknap1_1 | mknap1_7 | WEING1.DAT | PB7.DAT | mknapcb9-01 | mknapcb9-11 | mknapcb9-21 |
| MKP | LLM (B) | 11.65±5.26 | 13.69±4.53 | 4.51±2.11 | 4.93±0.56 | 5.05±2.14 | 8.14±4.88 | 1.5±0.26 |
| | LLM (E) | **0.00±0.00** | **0.00±0.00** | **1.83±1.83** | **1.96±0.6** | **1.08±0.8** | **2.23±0.93** | **0.9±0.45** |
| | Random (B) | 29.47±6.59 | 13.89±0.47 | 4.12±0.84 | 8.7±2.74 | 11.08±2.36 | 13.9±6.14 | 3.24±0.83 |
| | Random (E) | **0.00±0.00** | 4.56±0.24 | 4.31±0.82 | 8.38±3.52 | 6.41±2.62 | 6.14±3.3 | 4.67±0.46 |

From Table 7, we can observe the following points:

- In most case, the result from LLM selection is better than single heuristic and random selection.
- Selection from the evolved heuristics improved overall quality and reduced fluctuations in performance.
- Random selection performs worse than many single heuristic algorithms because poorly performing heuristics still have a chance of being selected.

We compare our evolution mothed with EoH and ReEvo by evolution nearest neighbor in TSP. Table 9 shows the average gap from evolved heurisitcs. EoH (GPT-4) and ReEvo (GPT-4) are reran on GPT-4 and ReEvo with default parameters, and result for ReEvo (GPT-3.5 Turbo) is sourced from ReEvo's paper.

The results in Table 8 show that our method (HeurAgenix , GPT-4) generally outperforms both EoH and ReEvo methods. The query count for EoH is fixed as 5 strategies * 10 population * 20 maximum iterations = 1000 queries in EoH (GPT-4). The query count for ReEvo is related to population size and evolution iterations with some fluctuations from LLM, and in this experiment the total number of queries for ReEvo (GPT-4) is 112. Our HeurAgenix has a query count that varies based on the number of training samples, perturbation success rate, and the number of bottlenecks identified per iteration, leading to some instability. In this experiment, the total number of queries for HeurAgenix is 228.

Table 8: TSP heuristic evolution experiment based on nearest neighbor. "-" indicates that the heuristics did not complete within the time limit (one hour). The best results are highlighted in **bold**. The nearest neighbor result is different from ReEvo because their implementation starts with a random selection while ours is fixed to the first node. The upper part is the test dataset in ReEvo, and the lower part is our data with large number of nodes.

| Instance | nearest neighbor | EoH (GPT-4) | ReEvo (GPT-3.5 Turbo) | ReEvo (GPT-4) | Ours (GPT-4) |
|---|---|---|---|---|---|
| ts225 | 20.41 | 18.33 | 6.6 | **6.02** | 8.5 |
| rat99 | 28.32 | 19.49 | 12.4 | 9.46 | **7.84** |
| rl1889 | 22.98 | 24.39 | 17.5 | - | **10.2** |
| u1817 | 25.92 | 22.28 | 16.6 | - | **11.08** |
| d1655 | 19.16 | 15.09 | 17.5 | - | **12.85** |
| bier127 | 14.76 | 14.63 | 10.8 | 12.49 | **10.2** |
| lin318 | 28.53 | 21.82 | 16.6 | 13.58 | **8.55** |
| eil51 | 19.95 | 9.86 | 6.5 | 7.38 | **6.1** |
| d493 | 19.04 | 22.03 | 13.4 | **11.3** | 18.2 |
| kroB100 | 31.69 | 9.84 | **12.2** | 12.66 | 12.88 |
| kroC100 | 26.4 | 16.71 | 15.9 | 14.17 | **9.49** |
| ch130 | 24.04 | 7.81 | **9.4** | 11.54 | 10.59 |
| pr299 | 24.28 | 19.41 | 20.6 | 19.89 | **11.4** |
| fl417 | 26.57 | 29.58 | 19.2 | 16.56 | **7.58** |
| d657 | 26 | 23.71 | 16 | 16.56 | **9.41** |
| kroA150 | 26.8 | 27.88 | 11.6 | 14.16 | **10.44** |
| fl1577 | 25.83 | 20.81 | 12.1 | - | **5.06** |
| u724 | 26.33 | 23.87 | 16.9 | 18.1 | **11.04** |
| pr264 | 18.09 | 17.6 | 16.8 | 15.32 | **11.73** |
| pr226 | 17.81 | 30.61 | 18 | 20.07 | **7.74** |
| pr439 | 22.44 | 22.89 | 19.3 | 18.4 | **7.73** |
| average gap | 23.59 | 19.94 | 14.57 | 13.98 | **9.93** |
| tsp225 | 28.35 | 25.11 | 18.32 | 9.33 | **5.31** |
| a280 | 22.41 | 17.56 | 12.49 | 15.61 | **10.00** |
| pcb442 | 22.03 | 29.56 | 16.85 | 15.86 | **11.99** |
| pa561 | 23.85 | 20.09 | 15.6 | 16 | **8.76** |
| gr666 | 24.67 | 19.1 | 21.91 | 21.91 | **13.72** |
| pr1002 | 27.82 | 26.28 | 21.87 | 19.96 | **9.74** |
| pr2392 | 21.99 | 22.86 | - | - | **12.91** |
| average gap | 24.45 | 22.94 | 17.84 | 16.44 | **10.35** |

We employ our evolved nearest neighbor generating init solution for GLS. Table 9 shows average gap.

Table 9: Comparison of TSP combination experiments with GLS using initial solutions from nearest neighbor (NN). NN(*) refers to the evolved nearest neighbor heuristic from HeurAgenix , and dist(*) refers to the updated distance matrix in EoH's paper. The best results are highlighted in **bold**. The upper part is the test dataset in EoH, and the lower part is our data with large number of nodes.

| Instance | NN | NN + GLS | NN(*) + GLS | NN + dist(*) + GLS | NN(*) + dist(*) + GLS |
|---|---|---|---|---|---|
| rd100 | 25.64 | 9.22 | 5.12 | **0.00** | **0.00** |
| pr124 | 17.39 | 2.44 | 1.45 | **0.00** | **0.00** |
| bier127 | 14.76 | 1.78 | 1.36 | 0.40 | **0.28** |
| kroA150 | 26.8 | 7.1 | 5.82 | **0.00** | **0.00** |
| u159 | 29.93 | 5.78 | 2.91 | **0.00** | **0.00** |
| kroB200 | 25.92 | 5.61 | 6.09 | **0.20** | 0.32 |
| average gap | 23.41 | 5.32 | 3.79 | **0.1** | **0.1** |
| tsp225 | 28.35 | 4.09 | 5.31 | 0.23 | **0.00** |
| a280 | 22.41 | 7.6 | 5.27 | 0.23 | **0.19** |
| pcb442 | 22.03 | 7.91 | 3.46 | 1.03 | **0.91** |
| pa561 | 23.85 | 5.79 | 5.36 | 3.4 | **2.71** |
| gr666 | 24.67 | 6.74 | 4.29 | 3.05 | **2.81** |
| pr1002 | 27.82 | 7.5 | 5.52 | 4.56 | **3.53** |
| pr2392 | 21.99 | 4.81 | 3.87 | 4.81 | **3.35** |
| average gap | 24.45 | 6.35 | 4.73 | 2.47 | **1.93** |

The experimental results in Table 9 show that our evolved nearest neighbor heuristic generally provides better performance when combined with GLS, compared to the standard nearest neighbor. Furthermore, the combination of our evolved nearest neighbor with the updated distance matrix from EoH and GLS also outperforms the corresponding standard nearest neighbor combination. This demonstrates that a better initial solution can enhance the effectiveness of hyper-heurisitics.

# F    COMMON STRATEGIES FOR HEURISTIC SELECTION

The strategies employed by the heuristic selection agent generally fall into four categories:

1. Select a constructive heuristic(e.g. nearest neighbor in TSP) to build an initial solution, then optimize it using improvement heuristics (e.g. 2-opt in TSP) until no further optimization is possible.

2. Try multiple constructive heuristics, observe feedback from the benchmark evaluation agent, select the best one, and then optimize the solution using improvement heuristics.

3. Switch different constructive and improvement heuristics based on different solution features during execution.

4. Try different combinations of constructive and improvement heuristics to find the optimal combination, and then run these fixed combinations.

Strategies 3 and 4 generally yield better results, indicating that real-time execution of improvement heuristics is more effective than first building and then optimizing the solution.

# G    DETAILED PROCESS AND PROMPT

In this section, we introduce the detailed process with prompt. {Placeholders} will be replaced with actual content content during program execution automatically.

**Standard Response Format**

Each prompt ends with a standardized response format, the `key` is a task-specific keyword recognizable by the next program, and we will omit in subsequent prompts for brevity.

**Standard Response Format**

```
The response format is very important.  For better communication,
please respond to me in this format:
***key:xxx***
Ensure there is no other content inside the ***, and analysis outside
*** are welcome.
If you have no information to provide, simply respond with ***None***.
```

**Background**

All tasks require background information, including problem description, data structure, code format, etc. Therefore, background are shared for varous tasks.

**Background**

```
I am working on Hyper-heuristics for Combinatorial Operation (CO)
problem.
In this conversation, I will introduce the problem and then framework
we have built now, you just remember this.
In next conversation, I will describe the challenges I'm encountering
and explore how we can collaborate to resolve them.

Currently, I am working on {problem} problem:
{problem_description}

To support different heuristic algorithms, I build the Solution and
Operator framework.
The Solution is designed as:
{solution_class}
Operator servers as a mechanism to modify solution, which enables the
application of heuristic algorithms.
To support heuristic algorithm, we have build the following operators:
{operator_class}

In pursuit of augmenting our heuristic algorithmic suite, we require
the following standardized heuristic function signature:
def heuristic(instance_data:  dict, solution_data:  dict,
algorithm_data:  dict, get_solution_data_function:  call) ->
tuple[TargetOperatorType, dict]:
The inputs are:
instance_data contains the instance data with:
{instance_data_introduction}
solution_data contains the solution data with:
{solution_data_introduction}
algorithm_data contains the hyper-parameters that necessary to control
algorithms.
get_solution_data_function is the function that receives the new
solution as input and return the state dictionary for new solution.
It will not modify the origin solution.
The outputs includes the operator that must be an instance of a
predefined target operator type and updated algorithm dict, which
contains new information for future work for both this or other
algorithm.

Please commit to memory the problem and our constructed framework.
```

## G.1 HEURISTIC GENERATION

**Generate From LLM**

> **Generate From LLM**
>
> I need your help to implement some basic heuristic for this problem {problem}.

**Learn from Paper:**

The detailed steps to learn from paper are as follows:

1. **Decompose Paper**: Decompose the paper into the abstract and various sections.

2. **Read Abstract**: The LLM reads the abstract to determine if the paper is relevant to the problem. If it deems the paper irrelevant or unsuitable for generating heuristics, the process is abandoned.

3. **Identify Interesting Sections**: If the abstract is relevant, the LLM identifies sections of interest, and we provide the content of these sections to the LLM.

4. **Evaluate And Generate**: Based on the section LLM chooses to 1) generates the heuristic; 2) abandons this paper; 3) continues to read additional sections.

**Reading Paper Abstract**

```
Here I will introduce a related paper for {problem}.

The title of this paper is:
{title}
The abstract of this paper is:
{abstract}

If you think we can not generate heuristic from this paper, we will
skip this paper.
If you think we can generate heuristic from this paper, we can work
in this way:  you provide the interested section and I provide the
content, until you think you rea ready to implement the code.

Please consider whether we can generate heuristic for {problem}:
1.  Consider whether this paper is related to {problem}.
2.  Consider whether this paper is suitable to generate heuristic, for
example some paper are related to this problem, but it is based on NN,
not heuristic, we have to ignore this paper.

Also remember we just generate one heuristic for this paper, so keep
focus on the best heuristic author claimed in paper.
```

**Read Paper Section**

```
Since this paper is suitable to generate heuristic for {problem}, we
start to read.
The previous section you are interested in is:  {last_interested_section}.
The content is:
{last_interested_content}

This is all sections in dict format:
{remaining_section_dict}

Please consider whether the read content are enough for you to generate
the heuristic for {problem}.
1.  If you think you are ready to implement the heuristic, respond to
me the heuristic name.
2.  If you think you need to read more, respond to me the heuristic
name.respond to me the interested sections.
3.  If you think we can not generate heuristic from this paper, respond
to me None.
Please select at most one section each time, and the section name
should align with provided dict.
Also to avoid the content is too large, we can start from leaf section.
```

**Transfer From Related Problem**

The detailed steps to transfer from related problem are as follows:

1. **Decompose New and Source Problems**: The LLM decomposes the new problem and source problems into components.

2. **Try to Match Components**: The LLM compares the components of the new problem with those of known problems to identify if heuristics from these problems can be leveraged.

3. **Read Source Heuristics**: If heuristics from known problems can be leveraged, the LLM reads the heuristics from these problems.

4. **Evaluate And Transfer**: For each heuristic, if the LLM determines it can be transferred, it translates the components and begins the transfer process; otherwise, skip this heuristic.

**Reference Problem**

```
This problem is a new classical problems, we can reference from some
classical problems.
We have already studied the following problems:
{studied_problems}

Please tell me which of these questions are relevant to our current
research.
```

**Mapping Component In Problem**

```
Now, try to analysis the similarities between {referenced_problem} and
this new problem {problem}

this is introduction for {referenced_problem}:
{referenced_problem_description}

Now I hope to decompose these 2 problems, find the similarities between
{referenced_problem} and this new problem {problem}, and mapping some
components.
```

**Reference Heuristic**

```
OK. Now let's review the all heuristic we have built for
{referenced_problem}:
{candidate_heuristic_pool}

Tell me, which heuristics can be transfer into {problem}?
It can be transferred from a single heuristic or multiple heuristics.
```

**Mapping Component In Heuristic**

```
Now, we have already found the similarities between
{referenced_problem} and this new problem {problem}:
{similarities_in_problem}

To support {referenced_problem}, I build the Solution and Operator
framework.
The Solution is designed as:
{referenced_problem_solution_class}
Operator servers as a mechanism to modify solution, which enables the
application of heuristic algorithms.
To support heuristic algorithm, we have build the following operators:
{referenced_problem_operation_class}

This is the code for {referenced_heuristic}:
{referenced_heuristic_code}

instance_data in {referenced_heuristic} contains the instance data
for {referenced_problem} with:
{referenced_instance_data_introduction}
solution_data in {referenced_heuristic} contains the solution data for
{referenced_problem} with:
{referenced_solution_data_introduction}

Try to make up the similarities between {referenced_heuristic} and
this new problem {problem}.
If no more similarities, return me ***similarities:None***
```

**Transfer Heuristic**

```
Let's try to transfer {referenced_heuristic}.

First generate a new heuristic name for this new heuristic and also
a new detailed description to guide us how to get the new heuristic
description for {problem}.
Please consider the differences between {referenced_heuristic} and the
new problem that may lead to different algorithms.
By the way, the last 4 digits after last '_' are identifiers and we can
ignore in new_heuristic_name.
```

**Implement Code**

LLM generates the detailed heuristic design with some common reminders, including specified input/output data formats, required libraries, annotations, and edge case considerations, etc, and then translates the design into code.

## Implement Code

Based on previous discuss, we get the heuristic {heuristic_name}: {description}
Now please implement it in python function {function_name}.
To implement this heuristic function in python, please note:
1.  We can assume that Solution and all Operators are imported in "src.problems.{problem}.components".
2.  The operator type must be defined previously, do not create a new one.
3.  Never modify the instance_data, state_data and algorithm data.
4.  All hyper parameters in algorithm_data should be set a default value, and use as algorithm_data.get("xx", default_value).
5.  Any reasonable partial solution may be used as input, such as an empty solution.
6.  Comments in the code are very important.  They must clearly explain which data are required by the algorithm, how the algorithm proceeds, and under what circumstances it will not return any operator or will return an empty operator.  We hope that people can understand the principles and workflow of the algorithm clearly just by reading the comments, without needing to look at the code.
7.  The name of function must be {function_name}.
8.  No any omissions or placeholders, I'm just going to use the code.
9.  For the algorithm to update the algorithm_data, do not modify directly "algorithm_data["abc"] = 123", we should return operator, {"abc":  123}.
10.  For the circumstances that algorithm return empty operator, please return None, {}.
11.  Make the result must be valid.

## Detailed Heuristic Design

Before implementing the heuristic, we need to verify its feasibility. Therefore, we will first attempt to translate this description into rigorous detailed design.

Please note:
1.  The heuristic function yields an Operator, a construct intricately designed to manipulate Solution instances.f the goals of the heuristic do not align with the existing Solution structure, it will be necessary to modify the algorithm so that it is compatible with the current Solution classes.  In the event that such modifications prove impossible, we may need to consider discontinuing the use of the algorithm.
2.  The state and instance_data have been detailed previously.  It is essential to determine whether the heuristic's logic requires any additional information beyond what has been provided.  If the heuristic logic naturally requires more data, please indicate this by returning "reasonable_input:  we need xxx inputs" and we will halt the implementation.
3.  The type of returned operator that the algorithm can potentially yield have been enumerated above.  If the heuristic logic naturally leads to an operator type that is not listed, please indicate this by returning "reasonable_output:  we need xxx operator" and we will halt the implementation.
4.  Currently our framework only support the single tour solution, so the heuristic algorithm must works on this design.  We can not merge and fusion of two or more solutions to get a new solution.
5.  We must assume that operator will run on current solution outside heuristic algorithm.
Now let's consider the logic for {heuristic_name}:  {description}
This involves evaluating whether the algorithm's intrinsic logic can be expressed within our Solution, and Operator constructs without necessitating further data or operator types.

**Smoke Test**

The detailed steps for smoke test are as follows:

1. **LLM predicts heuristic output**: The LLM predicts the heuristic's output based on the detailed heuristic description and smoke data.

2. **Run heuristic in environment**: We set up the environment and run the heuristic in smoke data.

3. **Validation and adjustment**:

    (a) **Crash**: If the run fails, return the exception to the LLM to further adjust the code until it is correct or abandon the heuristic.

    (b) **Inconsistent Results**: If the run is successful but the results are inconsistent, return both the expected and actual results to the LLM, and further adjust until correct or abandon the heuristic.

    (c) **Successful Test**: If the run is successful and the results are consistent with expectations, the code passes the test.

---

**Smoke Test Expected Result**

```
To verify whether the code is correct, we conducted a smoke test.
This is the test data:
{smoke_instance_data}

We run the following operations:
{previous_operations}

The current solution are
{smoke_solution}
with detailed data
{smoke_solution_data}

First think about what the expected output is.
```

**Smoke Test Compare**

```
In fact we run the {function_name} once, and now we got the output
operation from {function_name}:
{output_result}

The updated solution are
{updated_smoke_solution}
with detailed data
{updated_smoke_solution_data}

Please compare with your expected result: {expected_result}
1.  If the result is aligned with your target output, respond to me
***python_code:correct*** and we will save the code and finish this
generation.
2.  If the result is not aligned with your target output and you can
not generate correct one, respond to me ***python_code:None*** and we
will stop this generation.
3.  If the result is not aligned with your target output and you can
fix up this issues, update the python code in previous format.
```

> **Smoke Test Crashed**
>
> ```
> To verify whether the code is correct, we conducted a smoke test.
> This is the test data:
> {smoke_instance_data}
>
> While executing {function_name} with the given dataset, the program
> encountered an error and crashed.  The following error message was
> displayed:
> {error_message}
>
> Please try to fix it.  1.  If you think this heuristic can not be
> implemented, respond to me ***python_code:None*** and we will stop
> this generation.
> 2.  If you can fix up this issues, please update the python code in
> previous format.
> ```

### G.2 SINGLE ROUND EVOLUTION

The detailed steps for single-round evolution are as follows:

1. **Generate Comparison Data**

   (a) **Run Heuristic**: Use the heuristic and training data to generate an initial solution as the original solution.

   (b) **Perturbation For Better Solution**: Continuously perturb the original solution until a better solution is found, or abandoned if no better solution is found.

2. **Identify bottlenecks**

   (a) **Decompose**: Decompose both solutions.

   (b) **Identify Bottlenecks**: LLM identifies differences and identifies core differences that potentially impact solution quality, marking them as potential bottlenecks.

3. **Validate Each bottleneck**

   (a) **Reproduce Scenario**: For each bottleneck, we reproduce the scenario before them independently.

   (b) **Propose Suggestion**: The LLM proposes suggestion to replace the bottleneck.

   (c) **Verify Suggestion**: We validate by replacing the bottlenecks with proposed suggestion to test the suggested alternatives.

   (d) **Raise Experience**: If performance improves, LLM try to summarize this case and extract the suggestion; otherwise, we skip.

4. **Update Heuristic**

**Compare Solution**

```
In this instance, I have developed a heuristic function, although its
performance has not reached a satisfactory level.  My goal is to learn
from case studies to improve and optimize this heuristic.  To achieve
this, I will provide the following:
1.  The heuristic function code.
2.  Test data for evaluation.
3.  Negative solution from heuristic function.
4.  Positive solution from external.

The function {function_name} is the heuristic function:
{function_code}

The instance data for this problem:
{instance_data}

Negative solution from {function_name}:
{negative_solution}

Positive solution from external:
{positive_solution}

Please based on the data and solution, compare the difference between
these two solution and list the difference.
```

**Decompose Solution**

```
Then we decompose the solution.

The positive solution leads {positive_result} with the following
trajectory:
{positive_trajectory}
The negative solution leads {negative_result} with the following
trajectory:
{negative_trajectory}

Now we hope to analysis in operation level why negative operations
leads to poor performance.

Please note:
1.  Some operations look different, but actually express the same
effect.
```

**Identify Bottleneck**

```
Now, we hope to pick out the bottleneck operations in negative
solution.
Please note:
1.  Some operations, although they appear different, are essentially
the same.
2.  Some operations may lead to solutions that look different but are
essentially the same.
3.  Some operations may cause changes to the solution but do not affect
the final cost; these are not considered bottlenecks.
4.  When an operation A is performed poorly, leading to a series of
subsequent operations experiencing issues, we consider the first
operation A to be a bottleneck.

Please remember that these results were produced by {function_name},
and we hope to use them to identify the weaknesses of {function_name}.
Combine the solution_difference and operation difference before, try to
find out the bottleneck operations ids.
The negative solution leads {negative_result} with the following
trajectory:
{negative_trajectory}
```

**Propose Operation**

```
Now focus on {bottleneck_operation_id}:  {bottleneck_operation}.

Do not forget the instance data for this problem:
{instance_data}

The state before {bottleneck_operation} is:
{solution_data}

Please consider whether there is better operations in step
{bottleneck_operation_id} than {bottleneck_operation}.
To analyze the operation, we must delve into the detail design that
underpin it in following aspects:
1.  How can we get this operations, we need to analysis and calculate
to get this operation.
2.  Why this operation is superior.
3.  Examine the commonality of this phenomenon and identify any
specific conditions under which this operation is particularly suitable
or optimal, including instance data's conditions or current state's
conditions.
```

**Extract Suggestion**

```
To evaluate the validity of your suggestion, we keep the operations
before step {bottleneck_operation_id}, integrate {proposed_operation}
in step {bottleneck_operation_id} and applying the {function_name} for
remaining steps.  Now we got the update result

The updated result:  {proposed_solution} with {proposed_result}
{proposed_trajectory}

Compared with origin negative result from {function_name}:
{negative_solution} with {negative_result}
{negative_trajectory}

Your propose works well.

Now review the {function_name}:
{function_introduction}.
We hope to extract this into rule to get the suggestion for improvement
of {function_name}:
Please note:
1.  I believe that in most cases, our rule works in a scope of
applicability, that is, it is effective in certain circumstances.
Outside of this scope, we still maintain the original algorithm.
2.  The rule must be clear and calculate.  For example, choosing
operation A brings greater benefits in the form of rebates, but we
do not know how to measure future benefits.
3.  Rule must have nothing todo with current data.  It should be
general experience.

Combined previous calculate process:
{calculation_process}
And application scope:
{application_scope}
By the way, we believe no rule can works for all application scope,
sometimes it works and sometimes it may not work.  So application scope
is important.
Extract this analysis into rule to improve the {function_name}.
consider to raise suggestion:
1.  better selection
2.  better parameters
3.  insert more structure
4.  learn from other heuristics
```

**Sort Suggestion**

```
Now review the origin code {function_name}:
{function_code}

After analysis on between positive and negative solution, we have
already got some suggestions:
{suggestions}

We hope to apply these suggestions into nearest_neighbor heuristic,
while before to implement the code, we need to review and update the
suggestions:
1.  Some suggestions are similar or duplicated, we can merge them.
2.  Some suggestions conflict and we need to modify them.
3.  The application conditions of some suggestions are unreasonable, we
need to correct them.
4.  Some suggestions will bring too heavy calculation, we have to
optimize.
5.  We only need to keep the suggestions that have a greater impact and
are likely to be useful.

So based on these, please refine these suggestions with clear
conditions and sort them into heuristic code improve suggestion
consider to sort suggestion:
1.  better selection
2.  better parameters
3.  insert more structure
4.  learn from other heuristics
```

### G.3 GENERATE FEATURE EXTRACTOR

The detailed steps to generate feature extractor are as follows:

1. **Instance Feature Generation**: LLM lists the features of the instance data that characterized by:

   - **Distinct Characteristics**: Incorporating distinct attributes that help in clearly differentiating between various instances.

   - **Effective Representation**: Ensuring that the data representation is compact to reduce computational load.

2. **Solution Feature Generation**: LLM lists the features of the current soluton that characterized by:

   - **Characteristic Attributes**: Including unique attributes that facilitate the clear distinction between different stages of the solution process.

   - **Detailed Insights**: Maintaining a detailed enough representation to identify the specific characteristics of the current solution while being concise to ensure efficient processing.

   - **Comprehensive Evaluation**: Evaluating the current solution from various perspectives, such as the progress of the solution, its quality, and the status of the remaining data.

3. **Generate Feature Extractors**: LLM generates the feature extractors that ingests instance data and the current solution, then outputs the corresponding features.

4. **Smoke Test**: We validate the feature extractors by running with smoke test data and if the validation fails, the feature extractor functions are revised and updated.

**Instance Feature**

```
We aim to collaboratively create two distinct functions.  The first
function will be designed to methodically extract and distill features
from instance data.  The second function will focus on extracting
features that encompass both the characteristics and quality of current
solution.  These functions will be underpinned by rigorous statistical
analysis and domain-specific knowledge, ensuring they are both accurate
and relevant.

Let's begin by focusing on the features of the instance data for the
{problem}.
Instance data in the context of {problem} includes:
{instance_data_introduction}

In determining the optimal features (statistical data) to represent
instance features, we must adhere to the following criteria:
1.  The data representation should be succinct to minimize
computational load while retaining sufficient granularity to recognize
the feature of the problem and solution.
2.  It must incorporate unique attributes that aid in the clear
distinction between different instances.

Now, please tell me which features are best serve as instance features.
```

**Implement Instance Feature Code**

```
Let's go future.
Try to implement the get_instance_data_feature function in python:
def get_instance_data_feature(instance_data:  dict) -> dict

The input is instance_data, which contains the instance data with:
{instance_data_introduction}

The output is algo a dict, which contains the following features as
keys:  {instance_data_features}.

Please notes:
1.  Never modify the instance_data, solution_data and algorithm_data.
2.  The name of function must be get_instance_data_feature.
3.  No any omissions or placeholders, I'm just going to use the code.
4.  Comments in the code are very important.
```

**Solution Feature**

```
Then, let's focus on the features of the solution data for the
{problem}.
Instance data in the context of {problem} includes:
{instance_data_introduction}
Solution data in the context of {problem} includes:
{solution_data_introduction}

In determining the optimal features (statistical data) to represent
solution features and quality, we must adhere to the following
criteria:
1.  The data representation should be succinct to minimize
computational load while retaining sufficient granularity to recognize
the solution feature.
2.  It must incorporate unique attributes that aid in the clear
distinction between different solution stage.
3.  We need to evaluate the current status from multiple dimensions,
including the current progress of the solution, the quality of the
solution, the status of the remaining data, etc.

Now, please tell me which features are best serve as solution features.
```

**Implement Solution Feature Code**

```
Let's go future.
Try to implement the get_solution_data_feature function in python:
def get_solution_data_feature(instance_data:  dict, solution_data:
dict) -> dict

The input are instance_data and solution_data.
instance_data contains the instance data with:
{instance_data_introduction}
solution_data contains the solution data with:
{solution_data_introduction}

The output is algo a dict, which contains the following features as
keys:  {solution_data_features}.

Please notes:
1.  Never modify the instance_data, solution_data and algorithm_data.
2.  The name of function must be get_solution_data_feature.
3.  No any omissions or placeholders, I'm just going to use the code.
4.  Comments in the code are very important.
```

### G.4 HEURISTIC SELECTION

The detailed steps to select heuristics are as follows:

- **Input Information**: Instance features; Solution features; Heuristics description; Selection trajectory.

- **Chain of Thought (CoT) for heuristic selection in one query**:

    - **Analyze Problem Characteristics**: Based on the instance data features, analyze the problem's scale and characteristics to preliminarily assess the applicability of different heuristics.

    - **Evaluate the Current State**: Using the current solution features, evaluate the status and phase of the current solution to determine if further execution is necessary.

    - **Construct or Improve**: If further execution is needed, analyze whether to construct a new solution or improve the existing one.

    - **Narrow down Selection**: Based on the selection trajectory, identify potentially suitable heuristics and exclude those likely to result in poor performance.

    - **Assess Potential Heuristics**: Review the performance of the potential heuristics from tracjectory.

    - **Make Final Decision**: Using the heuristic descriptions, select the most appropriate heuristic.

- **Final decision**: Select heuristic, set parameters and execution step.

**Heuristic Pool**

We have already implement the following heuristics.
These are the heuristics inb format:  heuristic_name(
parameter=default_value, ..):  introduction
{heuristic_pool_introduction}

Before we solve the actual problem, please try to analysis the
scenarios where each algorithm is applicable, and these scenarios are
best quantif

**Heuristic Selection**

The instance data with some heuristic values for this problem:
{instance_data_feature}
Note:  Some data are omitted due to space constraints.

The solution data some heuristic values for current stage:
{solution_data_feature}
Note:  Some data are omitted due to space constraints.

Before this discuss, we have already {discuss_round} rounds discuss
and the summary are:
{heuristic_traject}

Considerations for Next Steps
- Is the current data sufficient for decision-making?
- Is there a need to construct or refine the solution further?
- The last heuristic is:  {last_heuristic}.  How does {last_heuristic}
perform, and should we continue with it?
- How much steps should we run for next heuristic?

Decision Options:
We aim to incrementally construct an optimal solution by strategically
applying a set of heuristic algorithms.  Each heuristic, when applied,
contributes one operator to the evolving solution.  Here is the refined
process:
1.  I will present you with the initial data.  Your role will be
to evaluate this data and select the most appropriate heuristic
algorithm from our pool of heuristics.  This selection will be based
on predefined criteria and heuristic performance measures.
2.  I will then execute the chosen heuristic for a number of steps,
resulting in a partial solution.  Once this stage is complete, I will
provide you with the updated solution state.  Your task will be to
assess the progress and determine whether to:
- Continue with the same heuristic algorithm to further develop the
current solution, or
- Switch to a different heuristic algorithm from our pool to either
enhance the existing solution or take a new approach to the problem.

As the selection hyper-heuristic algorithm agent, your role is critical
in orchestrating the application of these heuristics to navigate
towards an improved or final solution.  Please familiarize yourself
with the available heuristic algorithms and the overall decision-making
pipeline.  Once I introduce the specific data for our problem, we will
collaborate to advance towards the solution.

# H  INTRODUCTION TO CLASSIC COMBINATORIAL OPTIMIZATION PROBLEMS

**Traveling Salesman Problem (TSP)**   seeks to determine the shortest possible route that visits a given set of cities exactly once and returns to the origin city, based on the distances between each pair of cities.

**Capacitated Vehicle Routing Problem (CVRP)** involves determining the most efficient routes for a fleet of vehicles to deliver goods to various locations, taking into account vehicle capacity constraints.

**Job Shop Scheduling Problem (JSSP)** involves scheduling a series of jobs, each comprising a sequence of operations, across different machines to optimize production efficiency. Each job must be processed on specific machines in a predetermined order.

**Max Cut Problem** aims to partition the vertices of a graph into two disjoint subsets such that the total weight of the edges between the two sets is maximized.

**Multidimensional Knapsack Problem (MKP)** aims to maximize the total profit of selected items, each with a given profit value, subject to multiple constraints on the cumulative resource consumption of the items.

