# OpenReview forum: "HeurAgenix: A Multi-Agent LLM-Based Paradigm for Adaptive Heuristic Evolution and Selection in Combinatorial Optimization"
_ICLR.cc/2025/Conference — ICLR 2025 Conference Withdrawn Submission_

### Official Review · Reviewer_y5Nk · 2024-10-19

**Soundness:** 2
**Presentation:** 2
**Contribution:** 2
**Rating:** 3
**Confidence:** 4

**Summary:**

This paper introduces HeurAgenix, a multi-agent LLM-based framework for adaptive heuristic evolution and selection in solving combinatorial optimization (CO) problems. The framework consists of four key agents: a heuristic generation agent, a heuristic evolution agent, a benchmark evolution agent, and a heuristic selection agent. Experiments are conducted using GPT-4 on various CO problems, including TSP, CVRP, JSSP, MaxCut, and MKP.

**Strengths:**

1. This paper first combines the multi-agent concept with LLMs-based heuristic generation in solving CO problems.
2. The experiments are generally extensive.

**Weaknesses:**

1. This paper appears to involve substantial engineering and manual prompt design, which limits its novelty. As a follow-up to works like Funsearch, EoH, and ReEvo, either innovative optimization paradigms or significant empirical performance gains are expected to meet the bar of top-tier ML conferences.
2. The proposed framework is complicated, with lots of components. It is unclear which part contributes most to the observed improvements. Ablation studies would greatly clarify the importance of each component.
3. The empirical results are underwhelming. In Figure 9, why `GLS+EoH` is significantly better than your `GLS+Ours`?
4. The presentation of the paper could be improved. For example, the figures are visually unappealing, and Page 6 is entirely filled with figures.

**Questions:**

1. How about the sensitivity of the proposed framework w.r.t. the underlying LLMs?

---

### Official Review · Reviewer_1XeY · 2024-10-20

**Soundness:** 2
**Presentation:** 2
**Contribution:** 2
**Rating:** 3
**Confidence:** 4

**Summary:**

This manuscript proposes HeurAgenix to generate, evolve, evaluate, and select heuristics for solving Combinatorial Optimization Problems (COPs).

**Strengths:**

This manuscript dynamically chooses the most appropriate heuristics for different COP instances.

**Weaknesses:**

W1: On Page 2, the authors state that "However, these approaches still rely heavily on existing approaches". Yet, on Page 3, they acknowledge that their method also depends on existing heuristics: “the heuristic generation agent generates heuristics from LLM’s internal knowledge, reference papers, or related problems’ heuristics”. Please explicitly articulate the key innovations of HeurAgenix compared to previous LLM-based approaches.

W2: The method description is purely verbal, lacking essential mathematical definitions and formulas. This omission makes it difficult for readers to grasp some key steps and hinders reproducibility. For example, what specific conditions determine when a single round of evolution should be performed versus multiple rounds?

W3: On Page 4, the authors state that "Due to a phenomenon known as hallucinations, directly using LLMs to generate heuristics for new problems often leads to incorrect heuristics." I disagree with this characterization. The term “hallucination” typically refers to instances where LLMs generate false or fabricated information. In this context, heuristics that are unexecutable or perform poorly should not be described as instances of "hallucination".  Please provide a more precise description of the specific issues when using LLMs to generate heuristics.

W4: The inclusion of results for EoH [1] and ReEvo [2] in Figure 8 but not in Figure 6 raises concerns about the consistency of comparative data.

W5: In Section 4, the experiments are conducted exclusively with GPT-4, which raises concerns about the robustness of HeurAgenix when applied to other LLMs. Please evaluate HeurAgenix using multiple LLMs, such as Llama3-70b.

W6: To my knowledge, several deep learning methods have been proposed for selecting heuristics across diverse COP instances (e.g., [3, 4]). Incorporating comparative experiments with these methods would demonstrate the effectiveness of HeurAgenix .

W7: This manuscript omits several important experimental details and leaves key concepts undefined. For instance, the authors do not explain the "cheapest insertion method," which is presented as a baseline in Figure 6. This manuscript contains numerous grammatical errors and incoherent sentences. For instance, on Page 3, the sentence: “Selection hyper-heuristics optimize by selecting the most suitable heuristic from a predefined set to adapt to the current problem scenario.” Additionally, on the same page, the term "AppendixG.2". Please check the content of the entire manuscript carefully!


[1]  Evolution of heuristics: Towards efficient automatic algorithm design using large language model. In International Conference on Machine Learning, 2024.

[2]  Large language models as hyper-heuristics for combinatorial optimization, arxiv, 2024.

[3] A novel reinforcement learning-based hyper-heuristic for heterogeneous vehicle routing problem. Computers & Industrial Engineering,  2021.

[4] Selecting meta-heuristics for solving vehicle routing problems with time windows via meta-learning. Expert Systems with Applications, 2019.

**Questions:**

Please refer to the weakness section.

---

### Official Review · Reviewer_N7xH · 2024-10-22

**Soundness:** 3
**Presentation:** 2
**Contribution:** 3
**Rating:** 5
**Confidence:** 4

**Summary:**

This work introduces a novel multi-agent framework, HeurAgenix, that leverages LLMs to solve COPs. HeurAgenix comprises four key agents: heuristic generation, evolution, evaluation, and selection. These agents utilize LLMs' capacities such as autonomous knowledge synthesis, dynamic adaptability, and decision-making. HeurAgenix outperforms state-of-the-art approaches by generating scalable, data-efficient heuristics for both classical CO tasks (e.g., the Traveling Salesman Problem) and novel ones (e.g., Dynamic Production Order Scheduling).

**Strengths:**

- An important research topic.
- Targeting an emerging research field as a timely addition.
- The introduction of a novel problem ensures meaningful evaluation.
- A principled agentic framework is preferred over prior manual LLM+EA designs.
- Promising empirical performance.

**Weaknesses:**

- Some parts of the method seem case-specific rather than principled:
  - How do LLMs learn from reference papers or transfer knowledge from other problems during heuristic generation? Do you have to manually provide the involved knowledge sources and instructions?
  - In DPOSP experiment, how does the agent know the transferability between DPOSP and CVRP?
  - In single-round evolution, how do you perturb the solution? Do you have to manually design a perturbation heuristic in advance to enable the agentic framework?

- Generating features and selecting heuristics at every solution step can lead to substantial latency. However, the solution efficiency of your framework is left undiscussed.

- The overall framework contains many subtle mechanisms that are not sufficiently validated. For example, is the LLM-based smoke test effective? Can and why can LLM select effective heuristics given various complicated features?

- Limitations of your agentic framework are not discussed. The cost and latency are obvious limitations.

- Section 4.3 lacks experimental details. How do you ensure a fair comparison against prior LLM-based HH? For example, during training, do you use the same number of heuristic evaluations for all methods? During inference, do you implement both constructive and improvement heuristics while the baselines only implement the former? If so, does it make sense to also consider comparing runtime?

- API costs and runtime of your method should be detailed.

- Presentation should be improved. E.g., vector graphic is preferred; line 243: Figure3 ->  Figure3; line 245: AppendixG.2 -> Appendix G.2.

**Questions:**

Please refer to the weaknesses.

---

### Official Review · Reviewer_XmiD · 2024-11-03

**Soundness:** 3
**Presentation:** 2
**Contribution:** 2
**Rating:** 5
**Confidence:** 5

**Summary:**

This paper presents an LLM-based multi-agent framework named HeurAgenix for automated heuristic design. HeurAgenix utilizes LLMs not only for generating heuristics but also for their evaluation and selection. Additionally, it incorporates insights from related academic papers and heuristics applied to similar problems.  The experiments are conducted on several combinatorial optimisation problems. The results show effectiveness and promise when compared to existing LLM-based heuristic search methods, such as EoH and ReEvo.

**Strengths:**

1.The introduction of new components, such as benchmark evaluation and heuristic selection, enhances both performance and efficiency. The efforts to extract knowledge from academic papers and related problems are interesting.
2. The results have been effectively demonstrated across a variety of combinatorial optimization problems.

**Weaknesses:**

1. The framework comprises multiple components, some of which require interaction with LLMs tailored to specific problem designs. Although this complex framework might improve performance, it could hinder the generalization across different heuristic design tasks and various problems. The authors have shown applications in some combinatorial optimization problems; however, it would be beneficial if the authors could clearly outline which components necessitate task-specific adjustments and designs, how these effective designs can be implemented, and the extent to which different designs affect performance.
2. It is recommended to perform an ablation study of each component within the same tasks to clearly ascertain their individual effectiveness.
3. The paper could benefit from additional clarifications on the methodology (e.g., heuristics generation from reference paper and related problems) and experimental setups to further validate the claimed effectiveness.

**Questions:**

1. How does each component impact performance? Which components require problem-specific knowledge, and to what extent do they influence the results?
2. How many LLM queries are utilized in each component? Please provide examples or estimated numbers for LLM queries used in each component (including the self-correctness and all the other subcomponents), rather than a total number.
3. How many independent runs are conducted for comparisons on all the tested tasks? Are they use the same intial heuristics? How robust is the performance across different runs? Which main component(s) do you think most significantly impact the robustness of the results and how do you resolve it?
4. What would be the outcome if the same initial heuristics were used for both the proposed method and the comparison methods, such as EoH and ReEvo?
5. During the heuristic generation stage, how are reference papers selected, are the papers automatically searched, selected, scaned, and summarized by LLMs, and how many reference papers are utilized for generating initial heuristics in each task?
6. How are related problems determined? Does it mean that we already use domain knowledge on determine these problems? How many related problems are considered for each task?
7. In real-world applications, where there are often few or no related papers and existing problems, how is the methodology adapted?
8. How is the performance of the proposed method on the online bin packing problem, which has been tested on both FunSearch and EoH?

---

### Official Review · Reviewer_ae5Y · 2024-11-04

**Soundness:** 2
**Presentation:** 2
**Contribution:** 3
**Rating:** 3
**Confidence:** 3

**Summary:**

The paper introduces a novel approach that uses a large language model (LLM) to implement heuristics for combinatorial optimization problems. The method consists of four main phases: heuristic generation, heuristic improvement, benchmark evaluation, and heuristic selection. In each phase (referred to as an “agent” in the paper), an LLM is used to guide the process. The authors test their approach on six combinatorial optimization problems and report strong performance.

**Strengths:**

- **Research direction**
The research direction - using LLMs to create or adapt heuristics for combinatorial optimization - is very interesting and innovative.
- **Novelty**:
The method is unique. While similar studies have recently been published, I consider them concurrent work in this new field.

**Weaknesses:**

1. **Reproducibility**:
A major concern is that the paper does not provide enough detail to make the method fully reproducible. Since the approach has many components, each one needs a clear, in-depth description. However, the paper only describes these components at a high level, making it very difficult, if not impossible, for others to reimplement the method based solely on the information provided. The authors should offer much more detail on each phase.
2. **Lack of Ablation Studies**:
Another significant issue is the lack of ablation experiments. The paper provides limited insights into the inner workings of the method, such as which components are the most critical or which hyperparameters exist and which are the most important. Including an ablation study would help clarify the contribution of each part of the approach and offer valuable guidance for tuning and improving the method.
3. **Limited Evaluation on New Problems**:
The authors only evaluate the method on one new problem (My understanding is that the other problems tested were also used during development of the method). Given the claim that the method can be easily adapted to new problems, it would be more convincing if the evaluation covered a wider variety of problems (e.g., 10–20 problems). Testing on more problems would support the idea that the method can truly learn heuristics for many different combinatorial problems, rather than just performing well on a few hand-selected examples. Considering the simplicity of the tested problems and that only one new problem was used, the paper feels more like an early case study. However, because this research direction is so new, this limitation is more of a minor weakness.
4. **Overstatement of Performance**:
The authors state in the abstract that their method “significantly outperforms state-of-the-art approaches.” This claim may be too strong, as the comparison is only against other LLM-based methods, not traditional optimization approaches from operations research. Furthermore, the comparison to other LLM-based methods is only done on one problem (the TSP), so even the claim of outperforming all LLM-based approaches is not fully substantiated.

**Questions:**

- What is the runtime of the learned approaches?

---

### Note · Authors · 2024-11-14

I have read and agree with the venue's withdrawal policy on behalf of myself and my co-authors.